A new small-bodied ornithopod (Dinosauria, Ornithischia) from a deep, high-energy Early Cretaceous river of the Australian–Antarctic rift system

http://orcid.org/0000-0001-6355-0331 Herne Matthew C. 1 ornithomatt@gmail.com
http://orcid.org/0000-0003-0342-9964 Tait Alan M. 2
http://orcid.org/0000-0003-2370-4046 Weisbecker Vera 1
Hall Michael 2
http://orcid.org/0000-0001-9218-8124 Nair Jay P. 1
Cleeland Michael 3
http://orcid.org/0000-0003-4097-8567 Salisbury Steven W. 1
1 School of Biological Sciences, The University of Queensland , Brisbane, QLD , Australia
2 School of Earth, Atmosphere and Environment, Monash University , Melbourne, VIC , Australia
3 Bunurong Environment Centre , Inverloch, VIC , Australia
Farke Andrew
Electronic publication date: 2018 Jan 11
Publication date: 2018
Volume: 5
Electronic Location ID: e4113
Received 2017 Aug 17; Accepted 2017 Nov 9
Copyright: © 2018 Herne et al.
Copyright year: 2018
Copyright holder: Herne et al.
License: This is an open access article distributed under the terms of the Creative Commons Attribution License, which permits unrestricted use, distribution, reproduction and adaptation in any medium and for any purpose provided that it is properly attributed. For attribution, the original author(s), title, publication source (PeerJ) and either DOI or URL of the article must be cited.
License URL: https://creativecommons.org/licenses/by/4.0/

Keywords: Dinosaur, Sedimentology, Taphonomy, Ornithopod, Gondwana, Australia–Antarctica, Systematics, Pathology, Palaeoecology, palaeontology

Funding: University of Queensland Postgraduate Research Scholarship Graduate School Travel Grant School of Biological Sciences Travel Grant This work was supported by the University of Queensland Postgraduate Research Scholarship, Graduate School Travel Grant and School of Biological Sciences Travel Grant to Matthew C. Herne. The funders had no role in study design, data collection and analysis, decision to publish, or preparation of the manuscript.

==============================
A new small-bodied ornithopod dinosaur, Diluvicursor pickeringi, gen. et sp. nov., is named from the lower Albian of the Eumeralla Formation in southeastern Australia and helps shed new light on the anatomy and diversity of Gondwanan ornithopods. Comprising an almost complete tail and partial lower right hindlimb, the holotype (NMV P221080) was deposited as a carcass or body-part in a log-filled scour near the base of a deep, high-energy river that incised a faunally rich, substantially forested riverine floodplain within the Australian–Antarctic rift graben. The deposit is termed the ‘Eric the Red West Sandstone.’ The holotype, interpreted as an older juvenile ∼1.2 m in total length, appears to have endured antemortem trauma to the pes. A referred, isolated posterior caudal vertebra (NMV P229456) from the holotype locality, suggests D. pickeringi grew to at least 2.3 m in length. D. pickeringi is characterised by 10 potential autapomorphies, among which dorsoventrally low neural arches and transversely broad caudal ribs on the anterior-most caudal vertebrae are a visually defining combination of features. These features suggest D. pickeringi had robust anterior caudal musculature and strong locomotor abilities. Another isolated anterior caudal vertebra (NMV P228342) from the same deposit, suggests that the fossil assemblage hosts at least two ornithopod taxa. D. pickeringi and two stratigraphically younger, indeterminate Eumeralla Formation ornithopods from Dinosaur Cove, NMV P185992/P185993 and NMV P186047, are closely related. However, the tail of D. pickeringi is far shorter than that of NMV P185992/P185993 and its pes more robust than that of NMV P186047. Preliminary cladistic analysis, utilising three existing datasets, failed to resolve D. pickeringi beyond a large polytomy of Ornithopoda. However, qualitative assessment of shared anatomical features suggest that the Eumeralla Formation ornithopods, South American Anabisetia saldiviai and Gasparinisaura cincosaltensis, Afro-Laurasian dryosaurids and possibly Antarctic Morrosaurus antarcticus share a close phylogenetic progenitor. Future phylogenetic analysis with improved data on Australian ornithopods will help to test these suggested affinities.

Introduction

Lower Cretaceous fossil localities along the south coast of Victoria, southeastern Australia, reveal a rich terrestrial biota that inhabited volcaniclastic river floodplains within the extensional rift system between Australia and Antarctica (Fig. 1; Fig. S1) (Rich & Rich, 1989; Willcox & Stagg, 1990; Dettmann et al., 1992; Rich & Vickers-Rich, 2000; Rich, Vickers-Rich & Gangloff, 2002). Among the diverse assemblage of terrestrial and aquatic tetrapods currently recognised from this region—temnospondyls, crocodyliforms, ornithischian and theropodan dinosaurs, multituberculate, monotreme and tribosphenic mammals, plesiosaurs, pterosaurs and chelonians—small-bodied, turkey- to rhea-sized ornithopod dinosaurs were especially abundant and diverse (Woodward, 1906; Flannery & Rich, 1981; Molnar, Flannery & Rich, 1981; Rich & Rich, 1989; Rich & Vickers-Rich, 1994; Currie, Vickers-Rich & Rich, 1996; Rich, Gangloff & Hammer, 1997; Warren, Rich & Vickers-Rich, 1997; Rich & Vickers-Rich, 1999, 2000; Rich, Vickers-Rich & Gangloff, 2002; Rich & Vickers-Rich, 2004; Kear, 2006; Smith et al., 2008; Close et al., 2009; Rich et al., 2009a, 2009b; Barrett et al., 2010; Benson et al., 2010; Herne, Nair & Salisbury, 2010; Barrett et al., 2011a; Benson et al., 2012; Fitzgerald et al., 2012).

Figure 1 Maps showing positions of localities and regional geological features relative to the city of Melbourne.

(A) Australia, indicating the Otway region (box). (B) Positions of coastal vertebrate body-fossil localities in the Eumeralla Formation, faulting and location of section ‘A-A’ (see Fig. 4). (C) Southern Victoria showing subsurface extent of basin systems (dashed lines), outcrop (dark shaded areas) and vertebrate fossil localities (following Bryan et al., 1997). Dashed arrows in (C) indicate the direction of palaeo-flow from contemporaneous volcanism on the eastern Australian Plate margin (see Fig. S1). Abbreviations: EF, Eumeralla Formation; Lat., latitude; Lon., longitude; TSg, Tyers Subgroup; WF, Wonthaggi Formation.

Three ornithopod taxa have been named from the upper Aptian–lower Albian deposits in Victoria. These taxa include Leaellynasaura amicagraphica Rich & Rich, 1989 and Atlascopcosaurus loadsi Rich & Rich, 1989 from the Eumeralla Formation in the Otway Basin and Qantassaurus intrepidus Rich & Vickers-Rich, 1999, from the Wonthaggi Formation in the Strzelecki Group of the Gippsland Basin (Figs. 1B and 1C). The holotypes of these three Victorian taxa consist solely of fragmentary cranial remains, and of these taxa, postcranial remains have only been assigned to L. amicagraphica (Rich & Rich, 1989; Rich & Vickers-Rich, 1999).

Postcranial assignments to L. amicagraphica have included the small partial postcranium NMV P185992/P185993, discovered at the L. amicagraphica holotype locality in 1987, and regarded as a scattered part of the holotype (Rich & Rich, 1989), and several isolated femora, referred to the same taxon based on features shared with NMV P185992 (Rich & Rich, 1989; Rich & Vickers-Rich, 1999; Rich, Galton & Vickers-Rich, 2010). A second partial postcranium, NMV P186047, discovered at the L. amicagraphica holotype locality in 1989, was assigned to the informal femoral taxon ‘Victorian Hypsilophodontid Femur Type 1’ (Rich & Rich, 1989; Gross, Rich & Vickers-Rich, 1993). However, femora referred to ‘Victorian Hypsilophodontid Femur Type 1’ were later reassigned to L. amicagraphica by Rich & Vickers-Rich (1999). More recently, Herne, Tait & Salisbury (2016) considered all postcranial materials previously referred to L. amicagraphica inconclusive. Several additional ornithopod femora from the Victorian localities were also assigned to either Fulgurotherium australe von Huene, 1932, an ornithopod taxon based on femoral remains from the Albian Griman Creek Formation at Lightning Ridge, New South Wales (Molnar & Galton, 1986), or alternatively, the informal Victorian femoral taxon ‘Victorian Hypsilophodontid Femur Type 2’ (Rich & Rich, 1989). Rich & Vickers-Rich (1999) later reassigned all femora of ‘Victorian Hypsilophodontid Femur Type 2’ to F. australe. However, Agnolin et al. (2010) later considered F. australe a nomen dubium.

Of the handful of vertebrate fossil localities in the Otway region (Fig. 1), the locality of Dinosaur Cove has been the most intensively excavated, including tunnelling into the sea-cliff (Rich & Vickers-Rich, 2000). The holotype of L. amicagraphica and the two partial postcranial skeletons NMV P185992/P185993 and NMV P186047 were discovered within close proximity to each other during tunnelling at Dinosaur Cove (Rich & Rich, 1989; Rich & Vickers-Rich, 2000; Herne, Tait & Salisbury, 2016). Other vertebrate fossils from the Otway region have been discovered eroding out of the coastal shore platforms, such as the fragmentary maxilla of the Atlascopcosaurus loadsi holotype (NMV P166409) from the locality of Point Lewis (Fig. 1) (Flannery & Rich, 1981; Rich & Rich, 1989). In 2005, vertebrate fossils were discovered eroding from the shore platform at a new fossil locality near Cape Otway that came to be known as ‘Eric the Red West’ (ETRW) (Rich et al., 2009b) (Figs. 1 and 2). A partial postcranium (NMV P221080) subsequently excavated at ETRW was reported by Rich et al. (2009b) as a possible ornithopod. Preliminary sedimentological observations also reported by Rich et al. (2009b), considered that the small fragmented dinosaur carcass (NMV P221080) recovered from the site had been buried in sediments of a fast-flowing river after becoming entangled in a ‘trap’ of plant debris that accumulated around an upright tree stump.

Figure 2 Fossil vertebrate locality of Eric the Red West.

Shore platform looking west, showing undulating erosive boundary (solid white line) between the top of the Anchor Sandstone (AS) and the base of the ETRW Sandstone (ES). White dashed lines indicate selected bedding surfaces. White scale in mid-ground (indicated by arrow) equals 1 m.

In this investigation, the new partial postcranium from ETRW (NMV P221080) will be described and its phylogenetic relationships assessed. Sedimentology of the locality and taphonomy of the fossil assemblage will be investigated, extending from which, new insight on the palaeoecology of this region is anticipated. The relative stratigraphic ranges of the fossil taxa important to this work will be compared, assisted by a structural geological restoration of the Eumeralla Formation in the region of interest.

Materials and Methods

Information relevant to the specimens examined and compared is provided in Table S1. Specimens described in this work (NMV P221080, NMV P228342 and NMV P229456) were excavated using a rock saw, plug-and-feathers, jackhammer and hammers-and-chisels (DD) and prepared using mechanical methods (L. Kool, MU and D. Pickering, MV). Computed tomographical (CT) scan data (model: Siemens Sensation 64) were provided courtesy of St Vincent’s Public Hospital, Melbourne. CT scans for the anterior caudal vertebrae, up to Ca 6, and the lower right hind limb (1,147 slices; slice thickness 400 μm; voxel size 293/293/400 μm; peak X-ray tube voltage 140 kV, X-ray tube current 210 mA) were 3D modeled in Mimics Suite 14 (Materialise, Leuven, Belgium). Owing to poor resolution, CT scans for caudal vertebrae from Ca 7 (1,002 slices; slice thickness 1,000 μm; voxel size 574/574/1,000 μm; peak X-ray tube voltage 120 kV, X-ray tube current 62 mA) were not modeled. However, the output was viewed in OSIRIX (Pixmeo SARL, Geneva, Switzerland), which provided additional anatomical information reported within the description. The DICOM files are accessible at Figshare (http://dx.doi.org/10.6084/m9.figshare.5467990). Measurements of the bones were obtained directly using vernier calipers and indirectly from scale bars in the photographic images and digital tools within the CT viewing software. Nomenclature for vertebral laminae and fossa detailed in Table 1 follow the criteria of Wilson (1999), Wilson et al. (2011), Wilson (2012) and Tschopp (2016). The phylogenetic position of NMV P221080 was assessed within the datasets of Boyd (2015), Dieudonné et al. (2016) and Han et al. (2017) using TNT 1.5 (Goloboff & Catalano, 2016; Goloboff, Farris & Nixon, 2003).

Table 1 Nomenclature of vertebral laminae and fossae.

Lamina or fossa	Abbreviation	Landmark 1 or bounding margin 1	Landmark 2 or bounding margin 2	
Anterior centrodiapophseal lamina	acdl	Anteroventral margin of transverse process	Dorsolateral margin of anterior centrum	
Centroprezygapophyseal fossa	cprf	Ventral margin of prdl	acdl or dorsolateral margin of anterior centrum	
Centroprezygapophyseal lamina	cprl	Ventral margin of prezygapophysis	Dorsolateral margin of anterior centrum	
Posterior centrodiapophyseal lamina	pcdl	Posteroventral margin of transverse process	Dorsolateral margin of posterior centrum	
Postzygodiapophyseal lamina	podl	Dorsoanterior margin of postzygapophysis	Dorsal surface of transverse process	
Postzygoprezygapophyseal lamina	pprl	Postzygapophysis	Prezygapophysis	
Prespinal lamina	prsl	Medial margin of tprl	Anterior summit of spinal process	
Prezygodiapophyseal lamina	prdl	Lateral margin of prezygapophysis	Anterodorsal surface of transverse process	
Spinal ridge	sr	Medial margin of tprl	Medial margin of paired postzygapophyses	
Spinodiapophyseal fossa	sdf	Lateral surface of spinal process	Medial surface of podl and or transverse process	
Spinopostzygapophyseal lamina	spol	Posterior margin of spinal process	Medial margin of postzygapophysis	
Spinopostzygapophyseal fossa	spof	Left spol	Right spol	
Spinoprezygapophyseal fossa	sprf	Right sprl	Left sprl	
Spinoprezygapophyseal lamina	sprl	Spinal process	Prezygapophysis	
transprezygapophyseal lamina	tprl	Left prezygapophysis	Right prezygapophysis	
Note:

Nomenclature following Wilson (1999), Wilson et al. (2011), Wilson (2012) and Tschopp (2016).

Tail length of NMV P221080 was estimated from the combined length of the caudal vertebral centra. However, although the tail is articulated, its preservation in a curled state made the lengths of the intervertebral spaces difficult to measure with certainty. For this reason, the original tail length was estimated from the combined centrum lengths with the addition of an intervertebral gap of 11% (using criteria in Hoffstetter & Gasc, 1969). Precaudal body length of NMV P221080 was subsequently estimated from the comparative relative lengths of the anterior-most caudal vertebrae, precaudal vertebrae and cranial length in Hypsilophodon foxii (using Galton, 1974). From these body proportions, a restoration of NMV P221080 was attempted.

The site was mapped using compass, clinometer and tape. The positions of the fossil vertebrate localities of interest in the Eumeralla Formation utilised Land Channel coordinates (Department of Environment, Land, Water and Planning, State Government of Victoria). A regional geological section was produced (M. Hall, 1997–2005, field observations), upon which the localities were positioned and a subsequent restoration of syndepositional faulting for the Aptian–Albian produced. From this restoration, the relative stratigraphic positions of the localities were revealed, from which, the stratigraphic ranges of the fossil taxa were compared.

Nomenclatural acts

The electronic version of this article in portable document format (PDF) will represent a published work according to the International Commission on Zoological Nomenclature (ICZN), and hence the new names contained in the electronic version are effectively published under that Code from the electronic edition alone. This published work and the nomenclatural acts it contains have been registered in ZooBank, the online registration system for the ICZN. The ZooBank LSIDs (Life Science Identifiers) can be resolved and the associated information viewed through any standard web browser by appending the LSID to the prefix http://zoobank.org/. The LSID for this publication is: urn:lsid:zoobank.org:pub:0ACF3BE9-8E2F-4FEA-94B9-E418BE912418. The online version of this work is archived and available from the following digital repositories: PeerJ, PubMed Central and CLOCKSS.

Geographical and Geological Context

Lower Cretaceous strata of the Eumeralla Formation, Otway Group, crop out in sea-cliff and shore platform exposures along the south coast of Victoria, southwest of Melbourne (Figs. 1 and 2) and the primary vertebrate body fossil localities are located on the coastal margin between Apollo Bay and Dinosaur Cove (Felton, 1997a, 1997b; Rich & Rich, 1989; Wagstaff & McEwan Mason, 1989; Wagstaff, Gallagher & Trainor, 2012). The predominantly volcaniclastic sediments were deposited as thick multistory sheet-flood and river channel complexes within the half-graben resulting from crustal extension during rifting between Australia and Antarctica (Willcox & Stagg, 1990; Bryan et al., 1997; Felton, 1997b; Norvick & Smith, 2001; Duddy, 2003) (Fig. S1). The sediments were sourced from a contemporaneous, high-stand volcanic arc, resulting from subduction of the southwestern oceanic Pacific Plate along the eastern margin of the continental Australian Plate (Fig. 1C; Fig. S1) (see Bryan et al., 1997, 2002; Bryan, 2007; Norvick et al., 2008; Matthews et al., 2015; Tucker et al., 2016). The volcaniclastic sediments discharged westward into the Australian–Antarctic rift system as well as inland Australia (Fig. 1C; Fig. S1). Within rivers of the Australian–Antarctic rift, minor input of quartzose grit and gravel, derived from Palaeozoic basement detritus shed from the rift margins intermixed with the volcaniclastic sediments (Felton, 1997b). These extrabasinal sediments form thin discontinuous lenses within the sand bodies that crop out between Apollo Bay and Cape Otway (Felton, 1997b)—the region within which the vertebrate fossil localities of ETRW, Point Franklin and Point Lewis are located—but not at Dinosaur Cove, west of Cape Otway (Figs. 1B, 1C, 3A and 3B).

Figure 3 Depositional features of the ETRW sandstone.

(A) Gritty conglomerate trough cross-bed comprising coarse sand, quartzose/metamorphic gravel/grit matrix, mudrock rip-up clasts, coalified/carbonised wood fragments and vertebrate fossils. (B) Stacked, large-scale, medium- to coarse-grained sandstone and matrix supported conglomerate trough cross-beds. (C) Western-most section of excavation looking northwest, showing compacted coalified/carbonised woody debris (the partial postcranium NMV P221080 was excavated in the region immediately to the left of the log indicated). (D) Upright coalified tree stump and root-ball (dark bluish-grey mudstone) hosted by a conglomerate filled trough near the channel base, overlain by large-scale trough cross-beds of a clearer medium- to coarse-grained sandstone (lighter greenish-grey sandstone) that have buried the top of the coalified stump. Abbreviations: ctf, conglomerate trough fill; g, gravel/grit; mc, mudrock clast; rb, root-ball; tcb, trough cross-bed; tm, trough margin; ts, tree stump. Scale bars in A–B equal 0.5 m. Tree log length in C, ∼5 m.

Fossil localities of the Eumeralla Formation fall within the Crybelosporites striatus spore–pollen zone of Helby, Morgan & Partridge (1987), the base of which is at the Aptian–Albian boundary (113 Ma, following the time-scale of Gradstein, Ogg & Schmitz, 2012). The top of the Crybelosporites striatus spore–pollen zone is presently unresolved (following Wagstaff & McEwan Mason, 1989; Wagstaff, Gallagher & Trainor, 2012), but potentially middle Albian (∼109.5 Ma) (following Korasidis et al., 2016) (Fig. S2). Palynological studies further indicate that the fossil localities northwest of Cape Otway, in particular Dinosaur Cove, are younger than the localities northeast of Cape Otway, up to Apollo Bay (following Felton, 1997a, 1997b; Korasidis et al., 2016), which includes ETRW, Point Franklin and Point Lewis (Fig. 1B; Fig. S2). However, more precise chronostratigraphic resolution of these localities has yet to be published.

The vertebrate fossil-bearing localities of interest to this investigation include Dinosaur Cove (38°48′25.2″S, 143°27′28.8″E), ETRW (38°51′19.4″S, 143°31′53.0″E, between Cape Otway and Point Franklin), Point Franklin (38°51′20.9″S, 143°33′14.4″E) and the holotype locality of Atlascopcosaurus loadsi near Point Lewis (38°50′23.3″S, 143°34′28.2″E). A palaeolatitudinal reconstruction of East Gondwana for the Aptian–Albian (∼113 Ma) using GPlates (Müller, Gurnis & Torsvik, 2012) (Fig. S1) places southern Victoria, in the region of ETRW, at 68.0°S, 134.0°E.

Regional tectonic history and relative stratigraphic positions of the Eumeralla Formation fossil vertebrate localities

Deposition of the Eumeralla Formation coincided with north–south directed continental extension between Australia and Antarctica (see Fig. S1). Northeast–southwest trending normal faults and region-wide thinning of strata towards the northwest, coincided with half-graben development and regional crustal sag through thermal subsidence (see Hall & Keetley, 2009). Following the cessation of the continental extension phase between Australia and Antarctica at ∼95 Ma, rapid mid-Miocene to late-Pliocene oceanic plate divergence between these landmasses likely caused northwest–southeast crustal compression, resulting in folding and the inversion of normal faults from the Early Cretaceous (Veevers, Powell & Roots, 1991; Felton, 1992; Hall & Keetley, 2009). Although the fossil localities of the Eumeralla Formation are at the same relative level (i.e. shore level; Fig. 4A), their differences in age result from the complex tectonic history of compressive folding and faulting.

Figure 4 Schematic stratigraphic relationships of the Eumeralla Formation fossil vertebrate localities looking northeast along section ‘A–A’ (Fig. 1B).

(A) Present-day structural geological features. (B) Restored section for the early Albian (stratigraphic age estimates following Helby, Morgan & Partridge, 1987; Gradstein, Ogg & Schmitz, 2012; Wagstaff, Gallagher & Trainor, 2012; Korasidis et al., 2016). Stratigraphic zones ‘1’ and ‘2’ in (B) are arbitrary surfaces for reference between faulted blocks. Dashed line in (B) indicates present day coastal margin. Taxon abbreviations: C. paradoxa, Coptospora paradoxa; C. striatus, Crybelosporites striatus; C. hughesii, Cyclosporites hughesii; P. pannosus, Phimopollenites pannosus. Geological abbreviations: F, footwall; H, hanging wall. Horizontal scale approximate and vertical scale exaggerated.

As a result of regional structural deformation, the stratigraphic associations of the fossil vertebrate localities have been difficult to visualise in the field. Two northeast–southwest trending monoclinal faults, separated by ∼10 km, are observed in the region between Dinosaur Cove and Point Lewis (Figs. 1B and 4A). These include the Castle Cove Monoclinal Fault (strike 70°) to the south of Dinosaur Cove (Duddy, 1983; see Felton, 1992, fig. 2.4) and another fault north of Cape Otway (strike 45°), termed herein the ‘Cape Otway Monoclinal Fault’ (Duddy’s, 1983, ‘Cape Otway Anticline;’ see Felton, 1992, fig. 2.4). A further northeast–southwest trending fault located parallel to the coast borders the Torquay Sub-basin (Robertson et al., 1978; Felton, 1992; Hall & Keetley, 2009). These faults result in three main blocks (blocks ‘A,’ ‘B’ and ‘C;’ Fig. 4A) with the hinges of the asymmetric anticlines occurring on the hanging blocks, immediately northwest of the faults (Fig. 4A). Dinosaur Cove (dip 11–20°, az. 357°) is located on the northwest limb of the monocline on ‘block A’ (i.e. the hanging wall end of the block), while the three localities, ETRW (dip 12°, az. 346°), Point Franklin (dip 18°, az. 307°) and Point Lewis (dip 22°, az. 316°, 150 m southwest of Point Lewis; dip 27°, az. 300°, 200 m north of Point Lewis) are located on the northwest limb of the monocline on block ‘C’ (i.e. the footwall end of block ‘C’). The present-day dips at the fossil localities (Figs. 1B, 4A) are attributable to their positions on the long northwest limbs of the monoclines. The holotype locality of Atlascopcosaurus loadsi, near Point Lewis, is located 4.2 km northeast of ETRW and is stratigraphically lower than the latter (Fig. 4B) by a true stratigraphic thickness of ∼180 m.

The approximate stratigraphic relationships of the Lower Albian fossil localities in the Eumeralla Formation were further assessed within a preliminary structural geological restoration (Fig. 4B). On the restored section, Neogene aged reversal of the north–south trending, Aptian–Albian aged normal faults is removed and strata pinch towards their footwall ends—a typical feature of half-graben structures (Schlische, 1991). Dinosaur Cove, on block ‘A,’ is stratigraphically higher/younger than the fossil vertebrate localities of ETRW, Point Franklin and Point Lewis, on block ‘C.’ Thus, this restoration is consistent with palynological age estimates (Wagstaff & McEwan Mason, 1989; Felton, 1997b; Wagstaff, Gallagher & Trainor, 2012; Korasidis et al., 2016). At present, neither true stratigraphic thickness between Dinosaur Cove and the fossil vertebrate localities on block ‘C’ nor precise chronostratigraphic data for these localities are presently known. However, in the absence of precise chronostratigraphic data, the stratigraphic associations of the fossil localities of interest can at least be visualised from the restoration (Fig. 4B), and the stratigraphic ranges of the fossil vertebrate taxa within compared.

Sedimentology and taphonomy

Locality overview

The fossil vertebrate locality of ETRW is a shore platform exposure with low vertical relief (Fig. 2). However, local dip (14°) allows three distinct stratigraphic sequences to be tracked along the coast. The lowest unit observed in the region of the fossil site is termed the ‘Anchor Sandstone’ (Figs. 2 and 5), named for a ship’s anchor concreted onto rocks of this unit. The fossil-bearing unit of interest, termed the ‘ETRW Sandstone,’ erosively overlies the Anchor Sandstone (Figs. 2 and 5). The unit overlying the ETRW Sandstone is excluded from this present work.

Figure 5 Stratigraphic features of the Eumeralla Formation at the fossil locality of Eric the Red West.

(A) Stratigraphic profile. (B) Depositional features in the region of the western-most excavation. Abbreviations: c, conglomerate; m, mudstone; s, sandstone.

Anchor sandstone

Description: Only the top of the Anchor Sandstone is exposed at the fossil locality at low tide (Figs. 2 and 5). Owing to tilting, lower strata of the Anchor Sandstone are exposed on the shoreline to the southeast of the dig site. The unit fines upwards overall and is ∼30 m thick. The lower strata consist of large-scale cross-beds of medium to coarse-grained sandstone. The top beds comprise thinly laminated, interbedded, silty mudstone and wave-rippled, fine-grained sandstone, which passes up into a paleosol, comprising a pale-grey, unbedded mudstone, with a purplish-brown top layer.

Interpretation: Bedding of the Anchor Sandstone is indicative of a large channel sandbody that shows decreasing depositional energy from the unit base to its top. Prior to compaction, the deposit was >30 m thick, giving an approximate depth for the river channel. The lack of three-dimensional exposure of the unit inhibits conclusive assessment of the channel pattern. However, the bedding style suggests lateral accretion in a large meandering river channel (Allen, 1963, 1970; Walker, 1976). The thinly laminated, symmetrical rippled bedding at the top of the Anchor Sandstone (Figs. 2 and 5) formed from wind driven wave ripples in shallow water, such as in a shallow overbank lake (Nichols, 2009). A purplish coloured paleosol capping the rippled beds developed during a period of vegetation growth on the floodplain surface. Deposition of these upper beds would have been distant from the meandering channel (Kraus, 1999, p. 47).

ETRW sandstone

Description: The base of the ETRW Sandstone is scoured into the Anchor Sandstone forming an undulating contact with a relief of ∼0.5 m (Figs. 2 and 5). Tracking the bedding upwards from the unit base along the shoreline outcrop to the west of the fossil site indicates a total stratigraphic thickness of ∼25 m (Fig. 5A). The lower part of the ETRW Sandstone consists of overlapping, low-angled, large-scale trough cross-beds of medium- to coarse-grained sandstone (Figs. 2, 3 and 5). Some troughs are up to 10 m wide. The large-scale trough cross-beds extend upwards to at least half of the unit thickness. Many of the troughs in the basal few metres of the unit are scoured and infilled with, or floored by matrix-supported conglomerate, variably comprising medium to coarse sand grains, ‘grit’ (very coarse sand to small pebble size quartz and feldspar) with mica flakes, rounded mudstone rip-up clasts (typically up to 10 cm, and rarer clasts up to 25 cm), compacted, coalified/carbonized, river transported tree limbs/branches and logs (up to 1 m diameter and some up to 5 m in length) and tree stumps with root bases and attached soil (Fig. 3). The trough cross-beds pass up into climbing rippled beds of medium to fine-grained sandstone and interbedded, very fine-grained sandstone and siltstone layers at the unit top. Some layers show bioturbation (infilled burrows). Associated and isolated fossil vertebrate remains have been excavated from infilled scours within the basal 2 m of the ETRW Sandstone (Figs. 3 and 5).

Interpretation: The ETRW Sandstone is interpreted as a deep (>25 m) fluvial channel deposit with thinning-up of the bedding and fining-up of the grain-size indicating deposition by lateral accretion. However, conclusive interpretation of the channel pattern is inhibited by the lack of three-dimensional exposure. The large-scale trough cross-beds at the unit base (Figs. 3 and 5) are interpreted as the preserved parts of large migrating linguoid dunes on the channel floor (Simons, Richardson & Nordin, 1965; Walker, 1976). Trough cross-bed widths of up to 10 m indicate dunes of similarly large size within the channel (Simons, Richardson & Nordin, 1965; Rubin & McCulloch, 1980; Southard & Boguchwal, 1990; Boggs, 2001, p. 40–41). The thickness of the ETRW Sandstone indicates a meandering channel close to 1 km in width with a meander belt, if fully developed, nearing 10 km in width (based on criteria of Collinson, 1978). The discovery of isolated fossil bones and teeth in the deposit, provisionally identified as those of aquatic reptiles (see Rich, 2015), further supports the interpretation of a large permanent river.

The orientation of the troughs/scours, current-aligned logs and cross-bedding near the base of the unit indicates flow to the northwest (290°, based on present day coordinates; Fig. 5). Trough-shaped scours identified at the unit base, similar in size and orientation to those above the base, indicate scouring of the older Anchor Sandstone ahead of the migrating dune front. The flow rate of the river is suggested from two features. Firstly, flute marks identified at the unit base suggest upper regime flow of >1 m/s (Walker & Cant, 1984; Southard & Boguchwal, 1990) and secondly, at river depths of >20 m (i.e. the depth of the river that we expect formed the ETRW Sandstone), large-sized dunes form at flow velocities of ∼2.0 m/s (Rubin & McCulloch, 1980). The grit was potentially derived from the Palaeozoic basement of the rift margin (Felton, 1997b) and the mudrock clasts derived from the older, partly consolidated overbank sediments into which the river incised. The root bases of two current-aligned logs deposited near the partial postcranium (NMV P221080) are directed downstream (Figs. 3C and 5B). The current-aligned logs and tree stumps likely derive from cutbank collapse (e.g. Wood, Thomas & Visser, 1988; see also Seegets-Villiers, 2012, on the Wonthaggi Formation) and soil-derived mud retained around their root balls, suggest these heavy debris entered the channel close to the locality.

Coarse sediment in a river, along with tree debris, is typically mobilised during high stage flow (Walker, 1976). Peak migration of dunes similarly occurs during high stage flow, while peak aggradation, typically occurs during waning flow (Harms & Fahnestock, 1965; Allen, 1984). During high-stage flow in the river that formed the ETRW Sandstone, flow rate at the channel base would have been strong enough to mobilise a bedload mass of large waterlogged logs, tree stumps and branches. As the current slowed, movement of the logs and stumps likely halted. The grounded tree debris potentially formed obstructions, causing scouring and the entrapment of smaller plant debris as ‘logjams,’ which in turn may have entrapped smaller objects such as isolated ‘fresh’ and fossil bones and carcasses/body-parts, or caused the deposition of these objects in lee-side eddies.

Fossil context and taphonomic comments

The scours near the base of the ETRW Sandstone host a rich assemblage of isolated vertebrate bones (see also Rich et al., 2009b; Barrett et al., 2011a), among which, NMV P228342 and NMV P229456 (Fig. S3), two vertebrae of interest to this investigation, were excavated close to the partial postcranium (NMV P221080; Fig. 6). These two isolated vertebrae show minor breakage and erosion of their cortical surfaces (Fig. S3), suggesting they encountered only minor hydraulic reworking prior to their final deposition (Behrensmeyer, 1988). The partial postcranium NMV P221080 (Fig. 6) was discovered eroding out of the shore platform ∼3.0 m north of the shore platform edge (Figs. 2, 3 and 5B). The fossil is hosted by conglomerate extracted from a scour trough ∼1.2 m above the base of the unit (Figs. 3C, 5B; Fig. S4). The conglomerate additionally hosts compressed, coalified plant debris (Fig. S4), including large current-aligned logs (one immediately east of NMV P221080) and an upright tree stump (see also Rich et al., 2009b) with partial root ball attached (1 m north of NMV P221080; Figs. 3C, 3D and 5).

Figure 6 Partial postcranium, NMV P221080, assigned to the holotype of Diluvicursor pickeringi gen. et sp. nov., as prepared on five blocks of ETRW Sandstone.

(A) Specimen viewed from above, normal to the bedding. (B) Schematic. Abbreviations: as, astragalus; B #, host block number; Ca #, designated caudal vertebra and position; cal, calcaneum; fib, fibula; ha #, haemal arch/process and position; pd #, pedal digit number; tib, tibia. Image of NMV P221080, courtesy S. Poropat and Museums Victoria.

Compacted, coalified branches and finer plant fragments in host sediment of the ETRW Sandstone, surround the partial postcranium NMV P221080 (Fig. S4). Burial of NMV P221080 in coarse bedload, along with branches, sizable logs and tree stumps indicates the transportation and deposition of these remains during a period of substantial in-channel hydraulic flow. NMV P221080 likely entered the river from the floodplain upstream of the site as a carcass or body-part—the skeleton having been held together by soft tissues (muscles, skin, viscera, tendons and ligaments). Transportation and burial of NMV P221080 likely occurred over a short period of time, with destructive decay of the carcass/body-part and/or disarticulation by scavenging mitigated by rapid burial (Shipman, 1981; Behrensmeyer, 1982, 1988; Wood, Thomas & Visser, 1988). The anterior caudal vertebrae of NMV P221080 were preserved with their ventral surfaces oriented upwards. The haemal processes in this vertebral region were displaced from their life positions and laying flat in the bedding (Fig. 6; Fig. S4A). Displacement of these haemal arches further suggests that the soft tissues had been compacted by rapidly accumulating sediment. The carcass/body-part (NMV P221080) could have been deposited by eddy currents at the downstream edge of a woody mass of tree debris (‘logjam’), indicated by the current-aligned logs upstream of the fossil and the transported tree stump deposited close to the specimen (Figs. 3 and 5). NMV P221080 was likely to have been more complete when deposited, possibly a complete carcass, with loss of the original skeleton occurring in recent times from erosion of the shore platform (Figs. 2 and 5).

Systematic Palaeontology

ORNITHISCHIA Seeley, 1888

CERAPODA Sereno, 1986

ORNITHOPODA Marsh, 1881

Diluvicursor gen. nov. urn:lsid:zoobank.org:act:BB4925A8-A049-4569-9AF2-80B28E999279

Etymology: From the Latin ‘diluvi,’ for deluge or flood, in reference to the deep high-energy palaeo-river within which the type material was deposited and the palaeo-floodplain upon which the river extended, combined with the suffix ‘-cursor,’ from the Latin for runner.

Diagnosis: A turkey- to rhea-sized small-bodied ornithopod, differentiated from all other ornithopods by 10 potential autapomorphies: (1) dorsoventral height of the neural arch on the anterior-most caudal vertebrae (indicated at Ca 3), highly reduced and sub-equal to dorsoventral centrum height; (2) proximodistal length of the spinal process on the anterior caudal vertebrae (Ca 3–6), highly reduced and sub-equal to anteroposterior centrum length; (3) prezygapophysis on the anterior-most caudal vertebrae (up to Ca 5), horizontally oriented and located at the neural arch base, lateral to the neural canal; (4) tuberous process dorsally on the spinoprezygapophyseal lamina (sprl) of the anterior-most caudal vertebrae; (5) dorsoventrally narrowest part of the centrum on the posterior caudal vertebrae, distinctly offset posteriorly and embayed by a sulcus; (6) deep haemal groove present on all posterior caudal vertebrae; (7) triangular intervertebral process anteriorly on the centrum of the posterior-most caudal vertebrae incises a V-shaped notch at the posterior end of the adjoining centrum; (8) caudal ribs on the anterior-most caudal vertebrae (indicated at Ca 3) are transversely broad with the distance across the ribs ∼85% of total vertebral height (inclusive of haemal arch); (9) lateral distal tarsal embayed anteriorly by a sulcus for the calcaneum; and (10) pd IV-1 is strongly asymmetrical in dorsoplantar view (the proximal cotyle flares medially and the lateral edge is straight).

The taxon is further recognised by the combination of 12 shared features: (1) centrum on the middle caudal vertebrae deeply excavated by the haemal groove, as in Gasparinisaura cincosaltensis; (2) spinal process on the middle caudal vertebrae, steeply reclined to ∼30° from the dorsal plane, as in Valdosaurus canaliculatus; (3) distal end of the haemal process on the middle caudal vertebrae, anteroposteriorly expanded and distinct from the shaft, as in Gasparinisaura, Macrogryphosaurus gondwanicus, NMV P185992/P185993, NMV P186047, Parksosaurus warreni and Valdosaurus; (4) distal end of the haemal process on the middle caudal vertebrae, symmetrically expanded and disc-shaped, as in Parksosaurus; (5) distal end of haemal process on the posterior-most middle and posterior caudal vertebrae, asymmetrically expanded and boot-shaped, as in NMV P185992/P185993 and Camptosaurus dispar; (6) medial distal tarsal is thin, wavy and plate-like, quadrangular in shape and has a dorsoplantarly oriented groove on the proximal surface that extends between sulci on the dorsal and plantar margins, as in NMV P186047; (7) distal condyle on metatarsal (mt) I, plantomedially positioned relative to the diaphysis on mt II, as in Anabisetia saldiviai, NMV P185992/P185993 and NMV P1867047; (8) a hallux with relatively reduced dorsoplantar and transverse proportions (dorsoplantar heights of the distal condyle on mt I and pedal phalanx (pd) I-1 within 50% of the heights of the equivalent regions on pedal digit II), as in Anabisetia, Camptosaurus and NMV P186047; (9) pd I-1, asymmetric in dorsoplantar view, with the proximal cotyle flaring laterally while the medial edge is straight, as in NMV P185992/P185993; (10) plantar half of the diaphysis on mt II, transversely compressed to ≤50% of the equivalent region on mt III, as in Anabisetia, Dryosaurus altus, Dysalotosaurus lettowvorbecki, Gasparinisaura, Kangnasaurus coetzeei, NMV P186047, Morrosaurus antarcticus and Valdosaurus; (11) viewed proximally, mt II has a lunate profile (i.e. medially convex/laterally concave), as in Anabisetia, Gasparinisaura, Morrosaurus, NMV P186047 and the dryosaurids; and (12) viewed proximally, mt II has a keyhole-shaped profile as in Anabisetia, Eousdryosaurus nanohallucis and Gasparinisaura.

Diluvicursor pickeringi sp. nov. urn:lsid:zoobank.org:act:9E1765D7-756F-4CF2-A005-EC0B0BE996BA

Figures 6–27, 31, 33, 35, S3–S4; Tables 1–5

2009 Ornithopoda; Rich et al., p. 677.

2014 Ornithopoda; Herne, pp. 246–274.

Figure 7 Diluvicursor pickeringi gen. et sp. nov. holotype (NMV P221080), schematic restoration in left lateral view, showing preserved bones (light shading) and incomplete caudal vertebrae (outlined).

Abbreviations: as, astragalus; Ca #, designated caudal vertebral position; pd #, pedal digit number; tib, tibia.

Figure 8 Diluvicursor pickeringi gen. et sp. nov. holotype (NMV P221080), anterior caudal vertebrae.

A–B, Ca 3–6: (A) uncoated; and (B) NH4Cl coated, in ventral view. Abbreviations: Ca #, caudal vertebra and position; cdf, centrodiapophyseal fossa; cr, caudal rib; gv, groove; ha #, haemal arch/process and position; hg, haemal groove; ncs, neurocentral suture. Scale bar equals 50 mm.

Figure 9 Diluvicursor pickeringi gen. et sp. nov. holotype (NMV P221080), CT model of the anterior caudal vertebrae Ca 1–6.

A–E: (A) left lateral; (B) dorsal; (C) ventral; (D) anterior; and (E) posterior views. Short dashed lines are estimated bone margins. Abbreviations: Ca #, caudal vertebra and position; cr, caudal rib; dap, diapophysis; ha #, haemal arch/process and position; nc, neural canal; pprl?, uncertain postzygoprezygapophyseal lamina; pro?, uncertain processes/protuberance; prsl(p), prespinal lamina (and process); sp, spinal process; sprl(p), spinoprezygapophyseal lamina (and protuberance). Distances: ‘a,’ neural arch (=dorsal tip of spinal process to top of centrum or centre of the transverse process base); ‘b,’ vertebral height without haemal arch; ‘c,’ vertebral height including haemal arch; ‘d,’ transverse width across caudal ribs. Scale bars equal 50 mm.

Figure 10 Diluvicursor pickeringi gen. et sp. nov. holotype (NMV P221080), CT model of the anterior caudal vertebra Ca 4.

A–F: (A) left lateral; (B) right lateral; (C) ventral; (D) dorsal; (E) anterior; and (F) posterior views. Abbreviations: acdl, anterior centrodiapophseal lamina; cdf, centrodiapophseal fossa; cen, centrum; cprl, centroprezygapophyseal lamina; cr, caudal rib; fac, facet; lcf, laterocentral fossa; sprl, spinopostzygapophyseal lamina; nc, neural canal; ncs, neurocentral suture; pcdl, posterior centrodiapophyseal lamina; poz, postzygapophysis; pro, protuberance/process; prsl, prespinal lamina; prz, prezygapophysis; sp, spinal process; sprl(p), spinoprezygapophyseal lamina (and protuberance); spol, spinopostzygapophyseal lamina; tp, transverse process; tprl, transprezygapophyseal lamina. Scale bars equal 50 mm.

Figure 11 Diluvicursor pickeringi gen. et sp. nov. holotype (NMV P221080), anterior caudal vertebrae.

Ca 7–11, NH4Cl coated, in ventral view. Abbreviations: Ca #, caudal vertebra and position; cr, caudal rib; gv, groove; ha #, haemal arch and position; ha?, haemal arch with uncertain position; hg, haemal groove. Scale bar equals 50 mm.

Figure 12 Diluvicursor pickeringi gen. et sp. nov. holotype (NMV P221080), anterior to middle caudal vertebrae.

A–B, Ca 12–16: (A) uncoated; and (B) NH4Cl coated, in left lateral view. Abbreviations: Ca #, caudal vertebra and position; acdl, anterior centrodiapophyseal lamina; cr, caudal rib; ha, haemal arch/process; hg, haemal groove; lr(p), lateral ridge (and protuberance); ncs, neurocentral suture; pcdl, posterior centrodiapophyseal lamina; poz, postzygapophysis; prdl, prezygodiapophyseal lamina; prz, prezygapophysis; sp, spinal process. Scale increments in A equal 1 mm. Scale bar in B equals 50 mm.

Figure 13 Diluvicursor pickeringi gen. et sp. nov. holotype (NMV P221080), middle caudal vertebrae.

A–B, Ca 16–20: (A) uncoated; and (B) NH4Cl coated, in left lateral view. Abbreviations: Ca #, caudal vertebra and position; cpol, centropostzygapophyseal lamina; cprl, centroprezygapophyseal lamina; ha, haemal arch/process; hg, haemal groove; lr(p), lateral ridge (and protuberance); poz, postzygapophysis; pprl, postzygoprezygapophyseal lamina; prdl, prezygodiapophyseal lamina; prz, prezygapophysis; sdf, spinodiapophyseal fossa; sp, spinal process; sprf, spinoprezygapophyseal fossa. Scale increments in A equal 1 mm. Scale bar in B equals 10 mm.

Figure 14 Diluvicursor pickeringi gen. et sp. nov. holotype (NMV P221080), middle to posterior caudal vertebrae.

A–B, Ca 19–23: (A) uncoated; and (B) NH4Cl coated, in left lateral/lateroventral view. (C) Ventral view. Abbreviations: Ca #, caudal vertebra and position; ha, haemal arch/process; hg, haemal groove; prz, prezygapophysis; sp, spinal process. Scale increments in A equal 1 mm. Scale bar in B equals 50 mm. Scale bar in C equals 10 mm.

Figure 15 Diluvicursor pickeringi gen. et sp. nov. holotype (NMV P221080), middle to posterior caudal vertebrae.

A–B, Ca 21–25: (A) uncoated; and (B) NH4Cl coated, in left lateroventral view. Dashed arrows indicate change in centrum shape from quadrangular (box-like), at Ca 22, to hexagonal, at Ca 23. Specimen in lower image NH4Cl coated. Abbreviations: Ca #, caudal vertebra and position; ha, haemal arch/process; hg, haemal groove; lr(p), lateral ridge (and protuberance); prz, prezygapophysis; sp, spinal process; sul, sulcus on lateroventral fossa. Scale increments in A equal 1 mm. Scale bar in B equals 50 mm.

Figure 16 Diluvicursor pickeringi gen. et sp. nov. holotype (NMV P221080), posterior caudal vertebrae.

A–C: (A) Ca 27–38, NH4Cl coated; (B) Ca 27–30, with schematic; and (C) Ca 32–34, with schematic, in left lateral view. Abbreviations: Ca #, caudal vertebra and position; cen, centrum; cprf, centroprezygapophyseal fossa; ha, haemal arch/process; lr, lateral ridge; na, neural arch; ncs, neurocentral suture or location; poz, postzygapophysis; pprl, postzygoprezygapophyseal lamina; prdl, prezygodiapophyseal lamina; prz, prezygapophysis (l-, left; r-, right); sp, spinal process; sprf, spinoprezygapophyseal fossa; sr, spinal ridge; sul, sulcus on lateroventral fossa. Breakage indicated by cross-hatching. Scale increments in A–B equal 1 mm. Scale bar in C equals 10 mm.

Figure 17 Diluvicursor pickeringi gen. et sp. nov. holotype (NMV P221080), posterior-most caudal vertebrae.

A–B, Ca 34–38: (A) uncoated; and (B) NH4Cl coated with schematic, in left lateral view. (C) Ca 35–38 in ventral view. Abbreviations: Ca #, designated caudal vertebra and position; ha, haemal arch; hg, haemal groove; ivp, intervertebral processes; poz, postzygapophysis; pprl, postzygoprezygapophyseal lamina; prdl, prezygodiapophyseal lamina; prz, prezygapophysis; sr, spinal ridge; sp, spinal process; sul, sulcus on the lateroventral fossa. Scale bars equal 10 mm.

Figure 18 Diluvicursor pickeringi gen. et sp. nov., referred caudal vertebra, NMV P229456.

A–D, specimen with schematics in: (A) left lateroventral; (B) left dorsolateral; (C) ventral; and (D) right lateral views. Abbreviations: cen, centrum; cprl(f), centroprezygapophyseal lamina (and fossa); hg, haemal groove; lr, lateral ridge; poz, postzygapophysis; pprl, postzygoprezygapophyseal lamina; prdl, prezygodiapophyseal lamina; prz, prezygapophysis; sr, spinal ridge; sul, sulcus on the lateroventral fossa. Scale bar equals 50 mm.

Figure 19 Diluvicursor pickeringi gen. et sp. nov. holotype (NMV P221080), distal right crus, tarsus and pes.

A–B, distal tibia, fibula and proximal tarsus in anterior view and distal tarsus and pes in plantomedial view: (A) uncoated; and (B) NH4Cl coated. (C) Proximal tarsus in anteroventral view and distal tarsus and pes in proximo-plantomedial view. (D) Distal tibia and proximal tarsus in anterodistal view and distal tarsus and pes in plantomedial view, NH4Cl coated. Abbreviations: as, astragalus; cal, calcaneum; fib, fibula; fos, fossa; ldt, lateral distal tarsal; mdt, medial distal tarsal; mt #, metatarsal position; pd #, pedal digit number and phalanx position; tib, tibia; tub, tuberosity. Scale increments in B equal 1 mm. Scale bar in D equals 50 mm.

Figure 20 Diluvicursor pickeringi gen. et sp. nov. holotype (NMV P221080), CT restoration of the right distal crus, tarsus and pes.

A–B: (A) anterior/dorsal; and (B) plantar views. Abbreviations: as, astragalus; cal, calcaneum; fib, fibula; ldt, lateral distal tarsal; mdt, medial distal tarsal; pd, pedal digit number; tib, tibia. Scale bars equal 50 mm.

Figure 21 Diluvicursor pickeringi gen. et sp. nov. holotype (NMV P221080), CT model of the right distal crus and proximal tarsus.

A–G: (A) anterior; (B) posterior; (C) lateral; (D) medial; (E) distal, with proximal tarsus removed; (F) distal; and (G) proximal views. Abbreviations: as, astragalus; cal, calcaneum; cd, condyle; fib, fibula; imf, inter-malleolar fossa; lf, lateral fossa; lm, lateral malleolus; mm, medial malleolus; pmmr, posterior medial malleolar ridge; tib, tibia. Scale bar equals 50 mm.

Figure 22 Diluvicursor pickeringi gen. et sp. nov. holotype (NMV P221080), CT model of the right tarsus.

A–D, proximal tarsus in: (A) anterior; (B) posterior; (C) proximal; and (D) distal views. E–H, distal tarsus in: (E) proximal; (F) distal; (G) plantar; and (H) dorsal views. (I) Distal tarsus in proximal view, with metatarsus in situ. Dashed line in B indicates proximal margin of posterior ascending process. Abbreviations: as, astragalus; cal, calcaneum; aap, anterior ascending process; cal, calcaneum; fos, fossa; gv, groove; ldt, lateral distal tarsal; mdt, medial distal tarsal; mt #, metatarsal and position; mp, medial process; sul, sulcus; tub, tuberosity; pap, posterior ascending process. Scale bar equals 50 mm.

Figure 23 Diluvicursor pickeringi gen. et sp. nov. holotype (NMV P221080), CT model of the right pes.

A–B, pes with pedal digits I and V removed in: (A) dorsal; and (B) plantar views. C–D, partial metatarsus in: (C) proximal; and (D) distal views. Abbreviations: fos, fossa; gv mt I, groove for mt I; mt #, metatarsal position; pd #, pedal digit number and phalanx position. Scale bars equal 50 mm.

Figure 24 Diluvicursor pickeringi gen. et sp. nov. holotype (NMV P221080), CT model of right pedal digit I.

A–F, mt I in: (A) lateral; (B) plantar; (C) dorsal; (D) medial; (E) proximal; and (F) distal views. G–L, pd I-1 and pd I-2 in articulation in: (G) dorsal; (H) plantar; (I) medial; (J) lateral; (K) proximal; and (L) distal views. (M) pd I-1 in distal view. Abbreviations: abd, abductor surface/groove; add, adductor surface/groove; clg, collateral ligament groove/fossa; cot, cotyle; dia, diaphysis; ex, extensor groove or surface; fl, flexor groove or surface; pd #, pedal digit number and phalanx position; s mt II, surface for mt II. Scale bar equals 50 mm.

Figure 25 Diluvicursor pickeringi gen. et sp. nov. holotype (NMV P221080), CT models of right metatarsals II–III.

A–F, mt II in: (A) dorsal; (B) plantar; (C) lateral; (D) medial; (E) distal; and (F) proximal views. G–L, mt III in: (G) dorsal; (H) plantar; (I) lateral; (J) medial; (K) distal; and (L) proximal views. Abbreviations: abd, abductor surface/groove; add, adductor surface/groove; con, condyle; dia, diaphysis; ex, extensor groove or surface; fl, flexor groove or surface; fla, flange; fos, fossa; gv mt I, groove for metatarsal I; rid, ridge. Scale bar equals 50 mm.

Figure 26 Diluvicursor pickeringi gen. et sp. nov. holotype (NMV P221080), CT model of right pedal digit IV.

A–F, mt IV in: (A) dorsal; (B) plantar; (C) medial; (D) lateral; (E) proximal; and (F) distal views. G–L, pd IV-1 and pd IV-2 in: (G) dorsal; (H) plantar; (I) lateral; (J) medial; (K) distal; and (L) proximal views. Abbreviations: abd, abductor surface/groove; add, adductor surface/groove; clf, collateral ligament fossa; cot, cotyle; dia, diaphysis; dmc, dorsomedial condyle; ex, extensor groove or surface; fl, flexor groove or surface; fla, flange; fos, fossa; pd #, pedal digit number and phalanx position; plc, plantolateral condyle; rid, ridge. Scale bar equals 50 mm.

Figure 27 Diluvicursor pickeringi gen. et sp. nov. holotype (NMV P221080), CT model of right metatarsal V.

A–D: (A) plantar; (B) dorsal; (C) medial; and (D) lateral views. Abbreviations: dist, distal; prox, proximal. Scale bar equals 10 mm.

Table 2 Diluvicursor pickeringi gen. et sp. nov., holotype (NMV P221080), dimensions of caudal vertebrae.

Vertebra	Centrum APL	Centrum DVH	Centrum TW	Caudal ribs, total TW	Vertebral DVH (excluding haemal arch)	Haemal arch DVH	
Ca 1	Missing	–	–	–	–	21.7 inc	
Ca 2	Missing	–	–	–	–	30.3	
Ca 3	15.0 inc	10.0 a
10.0 p	9.6 a
10.0 p	41.5	19.5	23.3 inc	
Ca 4	15.2	10.2 a
9.4 p	9.5 a
9.5 p	39.0	20.5	28.0	
Ca 5	15.0	10.0 a
9.4 p	10.0 a
10.0 p	33.0	–	–	
Ca 6	10.5 inc	9.3 a	10.0 a	27.5 e	–	–	
Ca 7	9 inc	–	–	–	–	20.0	
Ca 8	14.0	–	9.0 a
9.0 p	–	–	–	
Ca 9	14.6	–	9.0 a
8.6 p	–	–	21.0	
Ca 10	15.0	–	9.2 a
9.0 p	–	–	18.0	
Ca 11	12.5 inc	–	–	–	–	–	
Ca 12	Missing	–	–	–	–	–	
Ca 13	13 inc	–	–	16.0 e	18.0	–	
Ca 14	14.3	8.0 a
9.2 p	12.0 e	–	20.0	13.0	
Ca 15	15.2	8.2 a
8.2 p	10.8 e	–	20.0	8.5	
Ca 16	13 inc	8.5 a
8.2 p	9.6 e	–	18.0	–	
Ca 17	17.0	8.0 a	10.4 e	–	15.2 inc	10.1	
Ca 18	16.6	8.1 a
8.0 p	10.4 e	–	16.0	11.1	
Ca 19	17.0	8.2 a
7.0 p	–	–	14.5	9.7	
Ca 20	17.0	7.0 a
6.0 p	–	–	–	–	
Ca 21	16.2	6.0 a
6.5 p	7.6 p	–	14.1	9.5	
Ca 22	16.0	6.0 a
6.3 p	7.0 p	–	13.2	9.0	
Ca 23	15.5	7.2 p	7.5 p	–	13.5	–	
Ca 24	15.8	7.0 p	7.0 p	–	13.5	–	
Ca 25	15.9	7.2 p	–	–	–	–	
Ca 26	Un-prepared	–	–	–	–	–	
Ca 27	16.0	4.6 p	–	–	10.0	–	
Ca 28	16.0	4.8 a
4.8 p	–	–	9.5	–	
Ca 29	15.5	5.0 a
4.8 p	–	–	9.0	–	
Ca 30	15.5	4.8 a
4.7 p	–	–	–	–	
Ca 31	12.8	4.5 a
4.0 p	–	–	–	–	
Ca 32	13.0	4.6 a
4.5 p	–	–	7.8	5.8	
Ca 33	11.9	3.5 a
3.0 p	–	–	6.0	5.8	
Ca 34	11.5	3.5 a
3.5 p	–	–	6.0	–	
Ca 35	11.5	3.0 a
3.5 p	–	–	6.1	–	
Ca 36	9.0	2.5 a
2.5 p	–	–	5.8	–	
Ca 37	8.0	2.5 a
2.5 p	–	–	4.5	–	
Ca 38	8.0 inc	2.5 a	–	–	3.5 inc	–	
Notes:

Dimensions in mm. Abbreviations: a, anterior end; APL, anteroposterior length; Ca #, caudal vertebra and position; DVH, dorsoventral height; e, estimated; inc, incomplete; p, posterior end; and TW, transverse width. Caudal vertebral sequence based on the first preserved haemal arch at the position designated Ca 1.

Table 3 Diluvicursor pickeringi gen. et sp. nov., holotype (NMV P221080), dimensions of the right crus.

Element	DTW	DAPW	NTWD	NAPWD	
Tibia	34.5	15.0	16.0	10.0	
Fibula	10.0	5.0	2.5	4.0	
Notes:

Dimensions in mm. Abbreviations: DAPW, distal anteroposterior width; DTW, distal transverse width; NAPWD, narrowest anteroposterior width of diaphysis; and NTWD, narrowest transverse width of diaphysis.

Table 4 Diluvicursor pickeringi gen. et sp. nov., holotype (NMV P221080), dimensions of right tarsus.

Element	GTW	GAPW	NAPW	GPDH	
Astragalus	27.0	16.0	10.0 (medial edge)	16.5	
Calcaneum	13.0	10.5	4.5 (medial process)	14.0	
Lateral distal tarsal	19.3	13.3	–	8.0	
Medial distal tarsal	19.0	14.6	–	4.0	
Notes:

Dimensions in mm. Abbreviations: GAPW, greatest anteroposterior width; GPDH, greatest proximodistal height; GTW, greatest transverse width; NAPW, narrowest anterioposterior width.

Table 5 Diluvicursor pickeringi gen. et sp. nov., holotype (NMV P221080), dimensions of right pes.

Element	PDL	PDPH	DDPH	PTW	DTW	
mt I	38.0	5.0	5.0	2.0	4.8	
mt II	58.5	20.0	10.5	8.5 d/4.0 p	8.0	
mt III	66.0	15.0	11.6	14.0 d/7.0 p	13.0	
mt IV	56.0	11.2	11.0	12.0	8.4	
mt V	21.5	3.9	–	1.9	–	
pd I-1	17.7	5.3	5.0	8.5	5.6	
pd I-2	8.0 inc	5.5	–	–	–	
pd II-1	–	10.5	–	8.5	–	
pd III-1	inc	–	–	–	–	
pd IV-1	17.1	–	–	11.2	7.9	
pd IV-2	inc	–	–	8.0	–	
Notes:

Dimensions in mm. Abbreviations: d, dorsal; DDPH, distal dorsoplantar height; DTW, distal transverse width; inc, incomplete; mt #, metatarsal position; p, plantar; pd #, pedal digit number and phalanx position; PDL, proximodistal length; PDPH, proximal dorsoplantar height; and PTW, proximal transverse width.

Distribution: Lower Cretaceous Australia.

Holotype: NMV P221080, partial postcranium, comprising an almost complete caudal vertebral series, the distal ends of the right tibia and fibula, complete right tarsus and partial right pes.

Holotype locality: Eric the Red West, ETRW Sandstone, lower Albian, Eumeralla Formation, Otway Group, southern Victoria.

Derivation of name: To acknowledge the significant contribution of David A. Pickering to Australian palaeontology and in memory of his passing during the production of this work.

Diagnosis: As for genus.

Referred material: NMV P229456, partial caudal vertebra from the holotype locality.

Description

Axial skeleton

Preservation and overview

Only caudal vertebrae are known from the holotypic axial skeleton with 38 vertebrae preserved in articulation (Figs. 6 and 7). The anterior-most preserved caudal vertebra (Ca) is represented by the haemal arch at the position designated ‘Ca 1,’ noting its true position within the vertebral sequence is unknown. The distal part of the neural spine is preserved at Ca 1 and the first preserved centrum at Ca 3 (Figs. 8 and 9). The ventral surfaces of Ca 3–11 are exposed and their dorsal surfaces are within the matrix (Figs. 9–11). CT imagery provides information on the neural arches from Ca 1 to 11 (Figs. 9 and 10, for Ca 1–6). The left and ventral surfaces of the caudal vertebrae posteriorly from Ca 13 are exposed and their right sides are within the matrix. The posterior portion of Ca 38 is missing. However, the left postzygapophyseal facet on Ca 38 indicates that additional caudal vertebrae would have been present in life. The caudal vertebral series is divided into three regions. The anterior region, identified by the presence of caudal ribs, extends from Ca 1 to 13. The middle and posterior regions are differentiated by a distinct change in centrum shape. The mid-caudal vertebrae extend from Ca 14 to 22 and the posterior caudal vertebrae from Ca 23 to 38 (Fig. 7). On the referred caudal vertebra NMV P229456 (Fig. S3), the left anterior and posterior lateroventral corners of the centrum are broken and the distal portion of the left prezygapophysis is missing. Unless indicated, the description is with respect to the holotype (NMV P221080). For nomenclature on the vertebral laminae and fossae, see Table 1.

Caudal vertebrae

Centra

The neurocentral sutures are clearly identified on the anterior-most vertebrae (Fig. 8A) and difficult to distinguish posterior to Ca 8. On the anterior caudal vertebrae, to at least Ca 8, the sutures lie ventral to the transverse processes. At Ca 10, the base of the transverse process is positioned on the neurocentral suture and at Ca 13 the transverse process appears to be located entirely on the centrum, ventral to the neurocentral suture (Fig. 12). The centra on the anterior-most caudal vertebrae have ovoid to U-shaped anterior and posterior ends (Figs. 8–10) and are elliptical in mid-transverse section. At Ca 3–6, the articulating surfaces of the centra are amphiplatyan (Figs. 9 and 10) and posteriorly to that position, are modestly amphicoelous (Figs. 11 and 12). The centra progressively decrease in dorsoventral height posteriorly along the tail and become anteroposteriorly longer towards the middle of the tail (Table 2). At Ca 17 to 18, the centra are up to 20% longer than those of the anterior caudal vertebrae. The centra remain axially elongate on the posterior caudal vertebrae. The anteroposterior lengths of the centra are marginally longer from Ca 17 to 30 than the centrum at Ca 4. Posteriorly from Ca 30, the centra become progressively shorter. The transverse shape of the centrum changes from ovoid on the anterior caudal vertebrae (i.e. Ca 3–13; Figs. 8–12) to quadrangular on the middle caudals (i.e. Ca 14–22; Figs. 12–15), to hexagonal on the posterior caudals (i.e. posteriorly from Ca 23; Fig. 15; Table 2).

The change in centrum shape between the middle and posterior caudal vertebrae results from the more ventral location of the lateral ridge on the latter vertebrae (Fig. 15). On the middle caudal vertebrae, a small protuberance is formed on the lateral ridge (Figs. 12–15). On vertebrae posteriorly from Ca 24, a small sulcus is formed on the lateroventral fossa of the centrum (Figs. 15–17) and offset posteriorly from the mid-point on the centrum. The sulcus is most strongly developed on vertebrae posteriorly from Ca 28 (Figs. 16 and 17). At Ca 35–38, unusual triangular processes developed at the anterior articular ends of the centra appear to incise corresponding notches at the posterior ends of the adjoining centra (Fig. 17). At Ca 3–11, haemal grooves are only shallowly developed (Figs. 8, 11), while on vertebrae posteriorly from Ca 14, the grooves more deeply excavate the centra (Figs. 12–15 and 17).

The centrum of the referred vertebra NMV P229456 (Fig. 18) is hexagonal in transverse section, has a posteriorly offset waist, although only shallowly developed, and is excavated ventrally by a haemal groove. The triangular anterior process, present on the posterior-most caudal vertebrae of the holotype, is lacking. NMV P229456 most resembles the caudal vertebrae on the holotype in the region of Ca 14–30. However, with an anteroposterior length of 26 mm, the centrum of NMV P229456 is approximately double the length of the centra in the region mentioned on the holotype.

Neural arches

At Ca 3–9, the spinal processes are straight, steeply reclined to ∼30° from the dorsal plane and have anteroposterior lengths sub-equal to the lengths of their centra (Ca 3–6, see Figs. 9 and 10; note, the neural arches on Ca 7–11 are not figured herein, but observed from CT output). At Ca 3 the dorsoventral height of the neural arch (measured from the dorsal tip of the spinal process to the centre of the transverse process; distance ‘a’ in Fig. 9A) is 44% of the total vertebral height, excluding the haemal arch (measured from the dorsal tip of the spinal process to the ventral-most margin of the centrum; distance ‘b’ in Fig. 9A) and 18% of total vertebral height including the haemal arch (distance ‘c’ in Fig. 9A). At Ca 3–9, the anterior and posterior margins of the spinal processes have constant anteroposterior widths (Ca 3–6, see Figs. 9 and 10). At the distal ends of these spinal processes, the dorsal tips are rounded, while their ventral tips are angular (Figs. 9 and 10). The shape of the spinal process abruptly changes at Ca 10 (observed from CT output). At Ca 10–19, the spinal processes are proximally constricted, in lateral view, and their distal ends expand to form paddle-shaped ends (Ca 12–19, see Figs. 12–14). At Ca 18–19, the distal ends of the processes are blunt, with distal expansion of the process greatest at Ca 18 (Figs. 13 and 14). On vertebrae posteriorly from Ca 10, the degree of distal expansion of the spinal processes progressively reduces and the distal ends regain a rounded profile (e.g. vertebrae posteriorly from Ca 12; Figs. 12–16). Spinal processes are developed up to Ca 27, after which point, a low spinal ridge is developed (Figs. 16 and 17) (Tables 1 and 2).

At Ca 3–5, the prespinal lamina (prsl) is prominently developed at the base of the spinal processes (Figs. 9 and 10). On vertebrae posterior to Ca 5, the prsl could be developed, but not identified in the CT output. At Ca 1–5, a thin flange-like process projects laterally from the left sides of the spinal processes near their distal ends (Figs. 9 and 10). The spinal processes on the middle caudal vertebrae remain straight and reclined at ∼30° from the dorsal plane (Figs. 12–15). However, in comparison to the anterior caudal vertebrae (Figs. 9 and 10), the spinal processes on the middle caudals are more elongate. As a result, relative to the heights of their centra, the neural arches on the middle caudal vertebrae are higher than those on the anterior caudals. At Ca 13, the dorsoventral height of the neural arch is ∼56% of the total vertebral height, excluding the haemal process, and at Ca 17–18, ∼65%.

At Ca 3–5, the pre- and postzygapophyses are horizontally oriented and located at the base of the neural arch, lateral to the neural canal (Figs. 9 and 10). The prezygapophyses extend only a short distance beyond the centrum. On vertebrae posteriorly from Ca 6, the pre- and postzygapophyses become more dorsally elevated relative to their neurocentral sutures and anterodorsally oriented. At Ca 10–11 (observed from CT output), the prezygapophyses are anterodorsally oriented to ∼30° from the dorsal plane and at Ca 13–15 (Fig. 12), the prezygapophyses are short, inclined to ∼40° from the dorsal plane and retracted posteriorly relative to the anterior ends of their centra. On vertebrae posteriorly from Ca 16, the prezygapophyses extend anteriorly beyond their centra and progressively become more horizontally oriented and dorsally convex (Figs. 13–17). At Ca 18–21, the prezygapophyses extend anteriorly from their centra by ∼25% of centrum length; at Ca 22–34, ∼30% of centrum length; and on vertebrae posteriorly from Ca 36, up to 50% of centrum length. On vertebrae posteriorly from Ca 23, the prezygapophyses are dorsoventrally expanded at their midpoint and rabbit-ear-shaped (Figs. 14–17).

On the anterior caudal vertebrae, the spinoprezygapophyseal lamina (sprl) connects the prezygapophysis to the lateral surface of the spinal process and demarcates the base of the prsl (e.g. Ca 4; Fig. 10). At Ca 3–5, a tuberous process is developed on the sprl, immediately posterior to the prezygapophysis (Figs. 9 and 10). The process on the sprl is weakly developed at Ca 6 and on the vertebrae posteriorly to that position, absent. At Ca 3–5, the transprezygapophyseal lamina (tprl) extends between the paired sprls (Figs. 9 and 10). On these vertebrae, the anterior edge of the tprl aligns with the anterior-most margin of protuberances on the prsls, as well as the posterior ends of the prezygapophyseal facets. In addition, the position of the tprl, dorsal to the neural canal, also corresponds to the dorsal margin of the prezygapophyses (Fig. 10).

The prezygodiapophyseal lamina (prdl) and postzygodiapophyseal lamina (podl) connect the pre and postzygapophyses to the transverse process, respectively (Fig. 10). On vertebrae posteriorly to Ca 13, the prdl merges with the lateral wall of the neural arch and the sprl and podl merge to form a single postzygoprezygapophyseal lamina (pprl; Figs. 12–17).

On the vertebrae posteriorly from Ca 17, a groove-like spinoprezygapophyseal fossa (sprf) is developed on the prezygapophyses, between the pprl and prdl (Figs. 13 and 16). The sprf is absent on the anterior caudal vertebrae and weakly developed on the anterior-most-middle caudals. On the anterior caudal vertebrae, the anterior and posterior centrodiapophyseal laminae (acdl and pcdl, respectively) connect the diapophysis to the base of the neural arch (Figs. 10 and 12). However, on the middle and posterior caudal vertebrae, the acdl and pcdl merge to form a continuous lateral ridge on the centrum ventral to the neurocentral suture (Figs. 12–17; see also ‘centra’ above). The centroprezygapophyseal lamina (cprl) and centropostzygapophyseal lamina (cpol) connect the pre and postzygapophyses to the base of the neural arch. The centroprezygapophyseal fossa (cprf) is formed laterally to the cprl. The cprf forms a weak depression on the anterior caudal vertebrae (Figs. 9 and 10), is well developed on the middle caudals (Figs. 12 and 13) and forms a narrow groove on the posterior caudals (Fig. 16). The cpol is indistinct on most of the caudal vertebrae and typically merges with the posterior margin of the pedicle.

The right prezygapophysis on the referred posterior caudal, NMV P229456 (Fig. 18), extends beyond the centrum by 30% of the centrum length, noting that the anterior-most tip of the right prezygapophysis could be missing. The prezygapophysis on NMV P229456 is dorsoventrally expanded at its mid-point and positioned close to the centrum, resulting in a narrow cprf (Figs. 18A and 18B). In lateroventral view (Fig. 18A), however, the cprf undercuts the ventral surface of the prdl—a feature also apparent on the posterior caudal vertebrae of the holotype. A spinal process is not developed on NMV P229456 and the postzygapophyses merge to form a median ridge (Fig. 18). The neural arch of NMV P229456 most resembles the posterior caudal vertebral positions Ca 28–32 on the holotype.

Transverse processes and caudal ribs

On the anterior caudal vertebrae, the transverse processes, upon which the caudal ribs attach, are laterally reduced and dorsoventrally thickened (Fig. 10). At Ca 3–5, the transverse processes are positioned centrally on their neural arches and at Ca 6–9, are more posteriorly positioned (Figs. 9–11). At Ca 10–13, the transverse processes regain central positions (Figs. 11 and 12). Transverse processes extend up to Ca 15. At Ca 14–15, the processes form short protuberances. Slight protuberances are also evident on the lateral ridge of the centrum on the vertebrae up to Ca 24 (Fig. 15).

The caudal ribs are fused to the diapophyseal facets on the transverse processes of the neural arches (Figs. 8–12). The transversely broadest distance across the caudal ribs, at Ca 3 (distance ‘d;’ Fig. 9B), is 85% of total vertebral height (distance ‘c;’ Fig. 9A; Table 2). On vertebrae posteriorly from Ca 3, the proximodistal widths of the ribs progressively decrease. In anteroposterior view, the caudal ribs of Ca 3–6 (Figs. 9D and 9E) are horizontally oriented and dorsally, shallowly concave. In dorsoventral view (Figs. 8–11), the caudal ribs at Ca 3 are orthogonal to the vertebral axis. At Ca 4–6 they are posterolaterally oriented, and at Ca 8, orthogonally oriented. The centrodiapophyseal fossa (cdf) excavates the proximoventral surface of the transverse process and extends ventrally onto the dorsolateral surface of the centrum (Figs. 8–10).

Haemal arches

The haemal arches are Y-shaped in anteroposterior view (see Fig. 11). The haemal canal is enclosed at its base and a median groove on the anterior surface extends from the proximal base onto the shaft of the haemal process (Figs. 8, 9 and 11). At Ca 3 (Fig. 9), the proximodistal height of the haemal arch is slightly less than three times the height of the neural arch (Table 2). On the vertebrae posteriorly from Ca 3, the proximodistal heights of the haemal arches progressively reduce. At Ca 15 (Fig. 12) the haemal arch is slightly shorter than neural arch height. At Ca 1–4, the haemal processes are proximodistally elongate and expand to a small degree at their distal ends (Figs. 8 and 9). At Ca 7–8, the haemal processes have small, paddle-shaped distal ends, and on the haemal arches posterior to Ca 9, the distal ends are anteroposteriorly expanded. On all of the haemal processes with anteroposteriorly expanded distal ends, the median shafts are distinct with parallel anteroposterior margins. At Ca 9–15, the distal ends are hatchet-shaped (Ca 14–15, see Fig. 12)—in reference to abrupt anteroposterior flaring of the process towards its distal end, its symmetrical form and abrupt truncation of the anteroposterior margins by a convex distal margin. At Ca 17–19 (Figs. 13 and 14), the processes are symmetrically disc-shaped—in reference to their rounded anterior, posterior and distal margins. A displaced, asymmetrical haemal arch lying ventral to Ca 19 (Fig. 14) is likely from Ca 20. On the vertebrae posteriorly from Ca 20, to at least Ca 23, the disc-shaped haemal processes become increasingly more posteriorly expanded, thus, asymmetrical (Figs. 14 and 15). Apart from fragments, the haemal arches are unknown between Ca 24 and Ca 31. At Ca 32–33, the haemal arches are distinctly boot-shaped (Fig. 16C)—in reference to distinct expansion and tapering of the processes towards their posterior ends and having relatively short, convex anterior ends. Haemal arches are developed up to the posterior-most vertebra preserved at Ca 38 (Fig. 17); however, the exact shapes of these processes are uncertain. The natural/correct orientations of the haemal arches are best observed at Ca 15–19 (Figs. 12–14). At these positions, the orientation of the haemal arches range from orthogonal to ∼80° from the dorsal plane. At Ca 21 and posteriorly to that position, the haemal processes are steeply reclined (Figs. 14A and 16C), which potentially occurred post-mortem.

Ossified tendons

From the CT imagery, elongate processes are observed on the left dorsolateral surfaces of the spinal processes at Ca 3–4 (Figs. 9 and 10). These features could be the fused remnants of ossified tendons. However, other than these features, ossified tendons are not apparent in the tail.

Appendicular skeleton

Preservation and overview

From the appendicular skeleton of the holotype, only the distal right crus, complete tarsus and partial right pes are known (Figs. 6, 7, 19 and 20). The pes is preserved in a state of postmortem hyperdorsiflexion (Fig. 19). Preparation has exposed the anterior and lateral surfaces of the crus, the anterior and ventral surfaces of the proximal tarsus and the plantar to lateromedial regions of the pes (Fig. 19). The metatarsals (mt) are imbricated (particularly mt II–III), most which potentially occurred during diagenetic compaction. A digital 3D model of the right distal hind limb is shown in Fig. 20 (noting that imbrication of the metatarsals has not been digitally adjusted). The calcaneum appears to have been displaced laterally from the astragalus by 2 mm. Pedal digit (pd) I is almost complete; however, the distal end of the ungual (pd I-2) is eroded. Phalanges pd II-1 and pd IV-1 are preserved and of these, only pd IV-1 is complete. Of the remaining phalanges, only the proximal portion of pd IV-2 is preserved.

Crus

Tibia

Viewed distally, the tibia is reniform (Fig. 21E). The narrowest transverse width of the preserved portion of the diaphysis is 47% that of transverse distal tibial width. The lateral malleolus is depressed distally relative to the medial malleolus (Figs. 21A and 21B) and ∼50% of the anteroposterior width of the medial malleolus. A shallow intermalleolar fossa is formed anteriorly (Figs. 21A and 21E). The posterior medial malleolar ridge is broad and shallowly rounded (Figs. 21B and 21E–21G; Table 3).

Fibula

The fibula is anterolaterally positioned relative to the tibia. The diaphysis is narrow with a D-shaped section and its anteromedial edge forms a thin crista that extends onto the distal condyle (Fig. 21A). The distal condyle is anteroposteriorly compressed, lunate in distal profile and flares towards its distal end where it would have articulated with the dorsolateral face of the calcaneum (Figs. 21A–21G)—noting that these two elements have been displaced by 5 mm on the holotype. Whether or not the fibula contacted the astragalus is uncertain (Table 3).

Tarsus

Astragalus

The astragalus and calcaneum are unfused and cap the distal end of the tibia forming a ginglymoid (saddle-shaped) surface (Figs. 21A and 21B). Viewed distally (Fig. 22D), the astragalus is sub-triangular—expanding medially and truncated laterally where it adjoins the calcaneum. A low tuberosity is present on the anteromedial face of the astragalus (Figs. 19D and 22A). The anterior ascending process on the astragalus is thin, centralized, transversely broad and the proximal (dorsal) margin, obtuse (Fig. 22A). A weak transverse fossa is formed anteriorly at the base of the anterior ascending process. The posterior ascending process (Figs. 22B and 22C) is thin and dorsally lower than the anterior process. In proximodistal view, the shallowly rounded profile of the posterior ascending process corresponds to the convex posterior surface on the medial malleolar ridge of the tibia (Figs. 21E, 21F, 22C and 22D; Table 4).

Calcaneum

The calcaneum is sub-circular in distal profile (Fig. 22D) with its transverse width slightly less than half that of the astragalus. A process on the mediodistal margin likely overlapped the adjoining lateral margin on the astragalus (Figs. 22C and 22D). The lateral surface forms a fossa (Fig. 21C) and the fibula likely articulated with the anteroproximal surface (Table 4).

Distal tarsus

The distal tarsus consists of the lateral and medial distal tarsals that rigidly cap the proximal end of the metatarsus (Figs. 19, 20 and 22E–22I). The medial distal tarsal, upon which the astragalus articulates, is a thin, wavy, quadrangular-shaped bony plate that caps metatarsals II and III. A shallow dorsoplantarly oriented groove is formed on the proximal surface (Fig. 22E) between sulci on the dorsal and posterior/plantar edges. This groove, however, does not correspond to the margin between metatarsals II and III. The convex distal surface on the medial distal tarsal is accommodated in a fossa shared proximally on metatarsals II and III (Figs. 22H and 22I). The lateral distal tarsal is wedge-shaped, tapering both laterally and anteriorly, is thicker than the medial and caps mt IV. The entire dorsolateral region of the lateral distal tarsal is embayed by a lunate fossa for the calcaneum (Figs. 22H and 22I). Mt V articulates with the plantar edge of the lateral distal tarsal (Table 4).

Pes

Metatarsus overview and surface orientations

The metatarsus is elliptical in proximal view and forms a compact, elongate, roughly cylindrical structure (Figs. 19, 20 and 23). The proximal surface of the metatarsus is angled to 30° from the transverse axis of the tarsus. In dorsal view, most of mt II is positioned plantar to mt III. Metatarsals I and V are positioned plantar to the rest of the metatarsus and not visible in dorsal view. However, to avoid complex directional terminology in the description and comparisons of the pedal elements, the orientations of the surfaces described are those that would be typically be used on a pes without substantial rotation of the digits. For example, the surfaces on mt II, which in reality are dorsomedially and plantomedially facing (see Fig. 23), are more simply described herein as dorsal and medial, respectively.

Pedal digit I (=hallux)

Mt I obliquely crosses the plantomedial face of mt II and is accommodated in a shallow groove on the latter (Figs. 19, 20B, 23 and 24). The proximal end of mt I forms a transversely compressed condyle (Figs. 24C–24E). The diaphysis is splint-like in its proximal half in the region adjoining mt II and becomes sub-triangular in section distally where it expands towards the distal condyle (Figs. 24E and 24F). The roughly spheroidal distal condyle is positioned plantar to mt II (Figs. 20, 23C and 23D). Viewed distally, the condyle forms a T-shaped profile with the head of the T facing medially (Figs. 24E and 24F). The grooves on the T-shaped condyle are interpreted as the extensor and flexor surfaces on the dorsal and plantar surfaces, respectively. The medial surface of the condyle, interpreted as the abductor surface, is smooth, while the adductor surface, laterally, is formed on the narrow condyle between the extensor and flexor grooves (Figs. 24A–24F). The distal condyle on the metatarsal is finely proportioned, with its dorsoplantar and transverse widths slightly less than 50% of those on the condyle of mt II (Table 5).

Two phalanges (pd I-1 and the ungual pd I-2; Figs. 19 and 24G–24M) are present. As preserved on the holotype, the proximodistal axis of pd I-1 is angled medially relative to the axis of the metatarsal. As a result, the phalanges of the hallux are oriented medially inwards (Figs. 19 and 20). In dorsoplantar view, pd I-1 is asymmetrical (Figs. 24G and 24H); the proximal end is laterally flared relative to the central axis of the diaphysis and the medial edge is straight. In mediolateral view (Figs. 24I and 24J), pd I-1 is dorsoplantarly compressed and the diaphysis recurves dorsally. Collateral ligament grooves are developed distally on the dorsal corners of pd I-1 (Figs. 24I, 24J and 24M). The plantar portion of the proximal ungual preserved on the holotype is eroded, while the dorsal surface, viewed distally, is rounded and sub-triangular (Fig. 24L; Table 5).

Pedal digit II

Mt II is elongate and closely articulates with mt III, which is accommodated in a fossa on the lateral surface that extends along the complete length of the bone (Figs. 19, 20B, 23 and 25A–25E). A fossa formed on the proximal surface of the metatarsal accommodates the medial distal tarsal and the medioproximal margin participates in the ankle joint (Figs. 20B, 25A–25D and 25F). The metatarsal forms a lunate, roughly keyhole-shaped profile in proximal view (Figs. 23C and 25F). Viewed mediolaterally, the metatarsal flares towards its proximal end, forming a fan-shaped profile (Figs. 25C and 25D). Surface bone on the proximal region of mt II is textured and rugose (Fig. 19). The plantar portion of the diaphysis on the metatarsal is transversely compressed over its length (Figs. 25B, 25F), with its width ∼33% that of mt III (Fig. 25). The ridge on the plantar surface of the metatarsal extends from the proximal end to the plantolateral corner of the distal condyle (Fig. 25B). The distal end of mt II is depressed plantarly relative to the diaphysis of mt III and forms a quadrangular-shaped, shallowly spheroidal articular condyle (Figs. 23D and 25E). The dorsoplantar depth of the distal condyle is greater than its transverse width. The flexor, extensor and abductor (medial) grooves are shallowly developed, while the adductor (lateral) groove is continuous with the lateral fossa for mt III (Figs. 25A–25E). Only the proximal portion of pd II-1 is preserved (Figs. 19A, 19B, 23A and 23B). The cotyle is rugose and envelops the distal condyle of mt II. The proximodistal axis of pd II-1 is mediodistally directed relative to the long axis of the metatarsal (Table 5).

Pedal digit III

The longest of the metatarsals, mt III closely adjoins mt II and mt IV (Figs. 19, 20 and 23). Viewed proximally, the metatarsal is roughly quadrangular in shape (Figs. 23C and 25L). A fossa on the proximal surface is continuous with mt II (Fig. 23C) and accommodates the convex distal surface of the medial distal tarsal. The dorsal surface of mt III is transversely broader than the plantar (Figs. 25G and 25H). In mediolateral view, the proximal end of the metatarsal is dorsoplantarly expanded, forming a T-shaped flange (Figs. 25I and 25J). Proximally, the bone is textured and rugose (Fig. 19). In dorsoplantar view, the metatarsal curves laterally outwards towards its distal end (Figs. 25G and 25H). The lateral and medial margins are shallowly concave and convex, respectively and the distal condyle recurves medially at the metaphysis. Viewed mediolaterally (Figs. 25I and 25J), the diaphysis of the metatarsal is dorsoplantarly compressed and shallowly bowed (dorsally concave–plantarly convex). The distal end recurves plantarly at the metaphysis to form a condyle that is spheroidal on the dorsal portion and centrally grooved plantarly, suggesting a ginglymoid joint (Figs. 25G–25K). Midway on the metatarsal, the diaphysis is rectangular in transverse section and dosoplantarly compressed. Collateral ligament grooves on the distal condyle of the metatarsal are shallowly developed. Only the proximal portion of pd III-1 is preserved (Figs. 19A, 19B, 20, 23A and 23B) and similarly to pd II-1, the cotyle forms a rugose, expanded flange. The proximodistal axis of pd III-1 is aligned with the mediodistally directed distal condyle on the metatarsal (Table 5).

Pedal digit IV

In mediolateral view, the proximal end of mt IV closely abuts mt III and the diaphysis abruptly expands both plantarly and laterally near its proximal end to form a triangular proximal flange (Figs. 19, 20, 23 and 26A–26D). A fossa formed on the proximal surface of the metatarsal, continuous with fossae on metatarsals II and III accommodates the lateral distal tarsal. The sulcus on the lateral distal tarsal for the calcaneum continues onto the proximal surface of the metatarsal and, as a result, mt IV participates in articulation with the calcaneum. Viewed dorsoplantarly, the metatarsal curves laterally outwards towards its distal end (Figs. 26A and 26B); the medial margin is convex where it abuts mt III. A shallow fossa on the medial surface of the diaphysis accommodates mt III (Fig. 26C). A narrow fossa on the plantar surface at the proximal end of the metatarsal (Figs. 19 and 26B) appears to be a natural feature, but alternatively could have resulted from diagenetic distortion. In transverse section, the diaphysis of the metatarsal is triangular in the region that adjoins mt III and becomes dorsoplantarly compressed and ovoid towards its distal end. Viewed plantarly, the proximal plantomedial edge of the metatarsal abutting mt III extends distally as an obliquely oriented ridge to connect to the prominent plantolateral process on the distal condyle (Figs. 26B and 26D–26F). The distal end of the metatarsal forms a spheroidal condyle that in distal view has a slanted, parallelogram-shaped profile (Fig. 26F) resulting from prominently developed mediodorsal and plantolateral processes. Flexor, extensor and adductor (medial) grooves are shallowly developed, while the abductor (lateral) groove is strongly developed. The axial length of mt IV is sub-equal that of mt II (Table 5).

As preserved, the axis of pd IV-1 is angled medially inwards relative to the distal axis of the metatarsal (Figs. 23A and 23B). The lateral region of the cotyle is either broken and missing or undeveloped (Figs. 26G, 26H and 26J). The plantar portion of the cotyle is split in the axial direction (Figs. 19A, 19B and 23B), which could be either pathological or taphonomic. Viewed dorsoplantarly, pd IV-1 is strongly asymmetrical (Figs. 26G and 26H). The proximal end flares medially relative to the diaphysis, while the lateral margin is straight (Figs. 26J and 26L). The cotyle forms a deep socket. The distal condyle is ginglymus and deep collateral ligament fossae are developed (Figs. 26I and 26J). Only the proximal portion of pd IV-2 is preserved, the cotyle of which closely fits the distal condyle on pd IV-1 (Figs. 26G–26K; Table 5).

Pedal digit V

Mt V is plantolaterally positioned relative to mt IV. The metatarsal is dorsoplantarly compressed and sickle-shaped (Fig. 27). The proximal end is thickened and rounded forming a condyle that articulates with the plantolateral margin of the lateral distal tarsal (Figs. 19, 20B and 22I). No phalanges are present (Table 5).

Comparisons

Caudal vertebrae

Increased centrum length on the caudal vertebrae of Diluvicursor towards the middle of the tail is shared with early ornithischians such as Agilisaurus louderbacki (Peng, 1992), Heterodontosaurus tucki (Santa Luca, 1980) and small-bodied ornithopods such as Jeholosaurus shangyuanensis (Han et al., 2012) and Valdosaurus (Barrett, 2016). Posterior offset of the dorsoventrally narrowest point on the lateroventral fossa of the centrum (i.e. the waist) on the posterior caudal vertebrae resembles morphology in D. lettowvorbecki (Janensch, 1955, table 12.26). However, the small sulcus on the ventrolateral fossa of the centrum in D. pickeringi (Figs. 15–17) appears unique, while noting this region is poorly described in many other taxa. Strongly developed haemal grooves on the middle caudal vertebrae of Diluvicursor (e.g. Figs. 14 and 17) resemble grooves at this location in Gasparinisaura (MCS Pv-1) and Heterodontosaurus (Santa Luca, 1980). Similarly to Diluvicursor, transversely broad haemal grooves are developed on the posterior caudal vertebrae of Gasparinisaura (MCS Pv-1), although grooves are only known up to Ca 27 in this taxon (the tail of MCS Pv-1 is not preserved posteriorly to that point). However, the haemal grooves in Gasparinisaura (MCS Pv-1) are not as deeply developed as those of Diluvicursor, and whether or not they persist to the posterior-most caudal vertebrae, as in Diluvicursor, is presently unknown. Deep haemal grooves have not been described on the posterior-most caudal vertebrae of any other ornithischian.

The dorsoventral heights of the neural arches on the anterior-most caudal vertebrae of D. pickeringi are lower than in all other ornithopods (Figs. 9 and 28). As in Diluvicursor, the spinal processes on the anterior caudal vertebrae of Anabisetia (PVPH-75 Cambiaso, 2007, fig. 105; M. C. Herne, 2008, personal observation), Dryosaurus (Galton, 1981), Dysalotosaurus (Janensch, 1955, pl. 13.5–6) and Valdosaurus (Barrett, 2016) are steeply reclined (30–45°), but differ from those of Diluvicursor in being comparatively lengthy. In most other ornithopods, the spinal processes on the anterior caudal vertebrae are comparatively upright and lengthy (e.g. Haya griva (Makovicky et al., 2011), Hypsilophodon (Galton, 1974), Mantellisaurus atherfieldensis (Norman, 1986), Othnielosaurus consors (Galton & Jensen, 1973), Parksosaurus (Parks, 1926, fig. 3, plate 11) (M. C. Herne, 2008, personal observation), Thescelosaurus neglectus (Gilmore, 1915) and Thescelosaurus sp. (CMN 8537, Sternberg, 1940)). As a result of the dorsoventrally low neural arches, the caudal ribs on the anterior-most caudal vertebrae of D. pickeringi are relatively broad. At Ca 3, the transverse width across the caudal ribs (distance ‘d’ in Fig. 9B) is ∼85% of total vertebral height (i.e. distance ‘c;’ Fig. 9A). In comparison, the broadest transverse width across the caudal ribs in H. foxii, at Ca 4 (NHMUK R196, using Galton, 1974, figs. 28, 29) is 55% of total vertebral height at that position.

Figure 28 Dorsoventral vertebral proportions on the anterior caudal vertebrae of selected ornithopods.

(A) Neural arch height ‘a’ (=height from dorsal tip of the spinal process to top of the centrum, or centre of transverse process base) relative to vertebral height ‘b’ (=vertebral height without haemal arch). (B) Neural arch height ‘a’ relative to vertebral height ‘c’ (=vertebral height including haemal arch). Distances ‘a’ and ‘b’ shown in Figs. 9 and 33 and distance ‘c’ shown in Fig. 9. Data sources, see Table S1. Tabulated data, vertebral positions and specimen numbers, see Table S2.

The steeply reclined condition of the spinal processes on the anterior caudal vertebrae of D. pickeringi continues to the middle caudal vertebrae. At the anterior-most middle caudal position (i.e. Ca 14), the total dorsoventral vertebral height is approximately double that of the centrum (see Fig. 29A). These dorsoventrally low proportions most closely resemble those of NMV P185992/P185993 and Gasparinisaura and to some extent Valdosaurus (Figs. 29B–29D) rather than taxa such as Hypsilophodon, Haya and Thescelosaurus sp. (CMN 8537, Sternberg, 1940), where dorsoventral vertebral heights are at least three times the length of their centra (Figs. 29E–29G).

Figure 29 Middle caudal vertebral profiles for selected ornithopods in left lateral view.

A–G: (A) Diluvicursor pickeringi gen. et sp. nov. holotype (NMV P221080), ∼Ca 14; (B) NMV P185992/NMV P185993, ∼Ca 14; (C) Gasparinisaura cincosaltensis, anterior-most posterior caudal (MUCPv-212, Coria & Salgado, 1996, fig. 4); (D) Valdosaurus canaliculatus, Ca 16 (Barrett, 2016, noting Ca 14 is transitional); (E) Hypsilophodon foxii, ∼Ca 13 (MNHUK R196, based on Hulke, 1882, pl. 74, fig. 13; following vertebral positions reported in Galton, 1974, figs. 28–29); (F) Haya griva, Ca 13 (following Makovicky et al., 2011, fig. 3, noting that caudal ribs persist along the entire vertebral series); (G) Thescelosaurus sp., Ca 13 (Sternberg, 1940, fig. 17). (H) Haemal process profile in NMV P186047, ∼Ca 14. Vertebral scales normalised for centrum length (‘f’) at Ca 14 on NMV P221080, with distances ‘a’ and ‘e’ based on the same vertebra, where ‘a’ equals neural arch height and ‘e’ equals vertebral height from the neurocentral suture to the ventral tip of the haemal process (i.e. ‘a’ plus ‘e’ equals total vertebral height, ‘c;’ Fig. 13). Abbreviations: cen, centrum; ha, haemal arch/process; ncs, neurocentral suture; sp, spinal process; tp, transverse process. Scale bars equal 10 mm.

Among ornithopods, the retracted prezygapophyses on the anterior-most middle caudal vertebrae of Diluvicursor resemble the condition in the dryosaurids Dryosaurus and Dysalotosaurus (Galton, 1981) and Thescelosaurus (Gilmore, 1915, fig. 7). An anteriorly prominent, tab-like prsl on the anterior caudal vertebrae (Figs. 9 and 10) resembles the prsl on the anterior caudal vertebrae of Camptosaurus (Gilmore, 1909, fig. 18), Eousdryosaurus (following Escaso et al., 2014), Haya (Makovicky et al., 2011, fig. 3), Thescelosaurus neglectus (Gilmore, 1915, fig. 6) and Ouranosaurus nigeriensis (Taquet, 1976). A thin prominent prsl on the caudal vertebrae of theropods, such as the abelisaurid Majungasaurus crenatissimus (O’Connor, 2007) suggests this feature could be plesiomorphic in dinosaurs, although variably expressed among taxa, or homoplastic.

The horizontally oriented caudal ribs in Diluvicursor (Figs. 9 and 10) resemble those in taxa such as Haya (Makovicky et al., 2011) and Hypsilophodon (Galton, 1974, figs. 28, 29) and differ from the posterodorsally directed ribs in Anabisetia (Cambiaso, 2007, p. 226; M. C. Herne, 2008, personal observation) and the dryosaurids, Dryosaurus (Galton, 1981), Dysalotosaurus (Janensch, 1955, pl. 13.4-11) and Valdosaurus (Barrett, 2016).

The hatchet-shaped haemal processes on the anterior-most middle caudal vertebrae of Diluvicursor (Ca 14–15) resemble the processes in similar positions on the caudal vertebrae of Gasparinisaura (MCS Pv-1), NMV P185992/P185993 and Valdosaurus (Barrett, 2016, fig. 4) (Figs. 29A, 29B and 29D). The disc-shaped, symmetrically expanded, haemal processes on all of the middle caudal vertebrae (Figs. 13–15) differ from the asymmetric, posteriorly expanded processes in Gasparinisaura (Coria & Salgado, 1996; MCS Pv-1, M. C. Herne, 2008, personal observation), Macrogryphosaurus (Calvo, Porfiri & Novas, 2007), NMV P185992/P185993 and NMV P186047 (Herne, 2014, fig. 9.6, 9.21–24), and in this aspect, resemble the symmetrically expanded middle caudal processes at Ca 7–15 in Parksosaurus (see also Parks, 1926, pp. 17–18). However, significantly differing from Diluvicursor, the processes in Parksosaurus are greatly expanded anteroposteriorly, with lengths sub-equal to their centra. Those of D. pickeringi are relatively small with lengths ∼40% of centrum length. The middle caudal haemal processes of Gasparinisaura (Coria & Salgado, 1996; MCS Pv-1, M. C. Herne, 2008, personal observation), Macrogryphosaurus (Calvo, Porfiri & Novas, 2007) and Parksosaurus differ from those of Diluvicursor, NMV P185992/P185993 and NMV P186047 in being more greatly expanded dorsoventrally. On the posterior caudal vertebrae of D. pickeringi (i.e. from Ca 23; Figs. 14–16), the haemal processes expand asymmetrically in the posterior direction and are boot-shaped, as in NMV P185992/P185993 (Herne, 2014, fig. 9.8). Boot-shaped haemal processes are also likely on the posterior caudal vertebrae of NMV P186047, while noting that the processes are not preserved posteriorly to Ca 17 (Herne, 2014). The boot-shaped haemal processes on the posterior caudal vertebrae of Diluvicursor, evident at Ca 32–33 (Fig. 16C), also resemble those on the posterior caudal vertebrae of Camptosaurus (‘C. browni’ Gilmore, 1909, fig. 19).

Longitudinal protuberances developed on the spinal processes of the anterior caudal vertebrae (Figs. 9 and 10) could be the fused remnants of ossified tendons, as in V. canaliculatus (Barrett, 2016). However, apart from these protuberances, ossified tendons are lacking in the tail of Diluvicursor, as in ornithopods such as Haya (Makovicky et al., 2011), Jeholosaurus (Han et al., 2012), NMV P185992/P185993 (Herne, 2009), Orodromeus makelai and Parksosaurus (see Brown et al., 2013), which differ from many other ornithopods, such as Hypsilophodon (Galton, 1974), Oryctodromeus cubicularis (Brown et al., 2013; Krumenacker, 2017) and Tenontosaurus tilletti (Forster, 1990), where ossified tendons ensheath the caudal vertebrae.

Caudal vertebral number

The total number of caudal vertebrae in D. pickeringi is unknown. However, utilising information from more complete small-bodied ornithopods, such as Haya, Hypsilophodon, Jeholosaurus and Thescelosaurus sp. (CMN 8537, Sternberg, 1940; Galton, 1974; Makovicky et al., 2011; Han et al., 2012), the number is estimated. Elongate, spine-like haemal processes typically present on the anterior caudal vertebrae of ornithopods help support the vertebral positions designated Ca 1–4 on the D. pickeringi holotype (Fig. 9). The anteroposterior length of the first preserved centrum on the holotype at the position designated Ca 3, is short relative to the anterior caudal vertebrae posteriorly to that position and the caudal ribs are transversely broader than the ribs posteriorly to Ca 3 (Fig. 9; Table 2). These features, apparent at Ca 3–4 in Hypsilophodon (following Galton, 1974), also support the position designated Ca 3 in the Diluvicursor holotype. The axial lengths of the caudal centra in D. pickeringi markedly decrease between the positions designated Ca 34 and Ca 38 on the holotype. The anteroposterior length of Ca 38 is 66% that of Ca 34 (Table 2), suggesting that Ca 38 is close to the terminal end of the tail. Although we cannot be certain, it seems unlikely that any more than ten vertebrae would have been originally present in the tail of the D. pickeringi holotype, posterior to Ca 38. The total number of caudal vertebrae in Diluvicursor was unlikely to have been greater than 50, as in Hypsilophodon, Thescelosaurus neglectus and Valdosaurus (Gilmore, 1915; Galton, 1974; Barrett, 2016).

Crus

Broad transverse expansion of the distal tibia in Diluvicursor (Figs. 21A, 21B and 30A) is typical for a neornithischian (e.g. Agilisaurus (Peng, 1992), Lesothosaurus diagnosticus (Thulborn, 1972) and ornithopods (Galton, 1974, 1981; Han et al., 2012)) and notably lacking in the heterodontosaurids, typified by Heterodontosaurus (Sereno, 2012, fig. 70; Galton, 2014). Compared to Diluvicursor, the distal ends on both the left and right tibiae of NMV P186047 are weakly expanded (Figs. 30B and 30C) and in this aspect, more comparable to Heterodontosaurus. Shallow posterior expression of the medial malleolar ridge on the tibia of Diluvicursor (Fig. 21E) seems unusual, with the ridge typically more pronounced in other ornithopods (e.g. Dysalotosaurus (Janensch, 1955, pl. 14.5b), Jeholosaurus (Han et al., 2012) and Mantellisaurus (Norman, 1986)).

Figure 30 Distal crura and tarsi of selected Eumeralla Formation ornithopods.

(A) Diluvicursor pickeringi gen. et sp. nov. holotype (NMV P221080), schematic right distal crus and proximal tarsus, in anterior view. B–C, NMV P186047, left distal crus and proximal tarsus showing: (B) image unaltered; and (C) reversed schematic, in anterior view. (D) Diluvicursor pickeringi gen. et sp. nov. holotype (NMV P221080), CT model of right distal tarsus, in proximal view. E–F, NMV P186047, left distal tarsus showing: (E) image unaltered; and (F) reversed schematic, in proximal view. Abbreviations: aap, anterior ascending process; as, astragalus; cal, calcaneum; dt?, distal tarsal, uncertain; fib, fibula; fos, fossa; gv, groove; ldt, lateral distal tarsal; mdt, medial distal tarsal; mm, medial malleolus; mt #, metatarsal number; tib, tibia. Scale bars equal 10 mm.

Proximal tarsus

The transversely broad, proximally obtuse, centrally positioned anterior ascending process on the astragalus of Diluvicursor (Fig. 22A), resembles the processes in Gasparinisaura (Salgado, Coria & Heredia, 1997, fig. 4.12), Dysalotosaurus (Janensch, 1955, pl. 14.5a), Talenkauen santacrucensis (Cambiaso, 2007, fig. 36A), Valdosaurus (Barrett et al., 2011b, fig. 8E) and possibly Notohypsilophodon comodorensis (Ibiricu et al., 2014, fig. 9G). The shape of the process in Diluvicursor is also is similar to those of Anabisetia, Dryosaurus and Muttaburrasaurus langdoni (Fig. S5). However, the shapes of the processes in the latter three taxa differ from that of D. pickeringi by having a well-developed fossa that borders the lateral margin (Fig. S5). The lateral margin on the process of D. pickeringi only forms a weak fossa (Fig. 22A). The anterior ascending process on the right astragalus of NMV P186047 (Figs. 30B and 30C) differs from that of Diluvicursor in being hook-shaped, as in Drinker nisti (Bakker et al., 1990, fig. 13) and Orodromeus (see Scheetz, 1999). The process in Jeholosaurus (Han et al., 2012) differs from that of Diluvicursor in being transversely narrow and tab-shaped, while the process in Hypsilophodon differs by forming a sharp cusp on the proximal margin (following Galton, 1974). Unlike, the anterior ascending processes on the astragali of Iguanodon bernissartensis (Norman, 1980, fig. 69a) and the rhabdodontids, Zalmoxes robustus and Zalmoxes shqiperorum (Weishampel et al., 2003) are medially offset.

The thin, rounded posterior margin on the astragalus of Diluvicursor, attributable to the shallowly developed medial malleolar ridge (Figs. 21F, 22C and 22D), contrasts with other ornithopods, where the posterior margin is typically more protrusive (e.g. Anabisetia, Dryosaurus (Galton, 1981, fig. 18f), Dysalotosaurus (Janensch, 1955, table 14: figs. 5a, b), Hypsilophodon (Hulke, 1882, pl. 80, figs. 5, 7), Muttaburrasaurus (Bartholomai & Molnar, 1981, fig. 10) and Tenontosaurus (Forster, 1990)). The low tuberosity on the anteromedial face of the astragalus in Diluvicursor (Figs. 19C, 19D and 22A) somewhat resembles the rugose feature described in Valdosaurus by Barrett et al. (2011b, fig. 8E).

Distal tarsus

The presence of two distal tarsals is typical for an ornithopod, although differing from Jeholosaurus and Orodromeus that possess three distal tarsals (Han et al., 2012). The thin, wavy, approximately quadrangular-shaped medial distal tarsal of D. pickeringi closely resembles that of NMV P186047, including the presence of a centrally positioned, dorsoplantarly oriented groove on the proximal surface that extends between sulci on the dorsal and plantar margins (Figs. 30D–30F).

Pes

The compact, elongate metatarsus of D. pickeringi, with a splint-like mt V and hallux retaining two phalanges (Figs. 19 and 20), is plesiomorphic for an ornithopod and typically present in non-dinosaurian avemetatarsalians (=avian-line archosaurs; Nesbitt, 2011; Becerra et al., 2016; see also Nesbitt et al., 2017). The approximately cylindrical-shaped metatarsus of Diluvicursor, resembles those of many other small-bodied ornithopod bipeds (e.g. Anabisetia (Coria & Calvo, 2002), Gasparinisaura (Salgado, Coria & Heredia, 1997), Hypsilophodon (Galton, 1974), Jeholosaurus (Han et al., 2012) and Orodromeus (Scheetz, 1999)). The oblique angle on the proximal margin of the metatarsus of Diluvicursor (Fig. 23) resembles the condition in Diluvicursor (Cambiaso, 2007, fig. 76A, Salgado, Coria & Heredia, 1997, fig. 5.4) and Orodromeus (Scheetz, 1999) and the early neornithischian Hexinlusaurus multidens (He & Cai, 1984, plate 4.4).

Among ornithopods, a finely proportioned pedal hallux, where the dorsoplantar and transverse proportions of the two phalanges (indicated by pd I-1), as well as the distal condyle on mt I, are reduced to within 60% of the equivalent regions on pedal digit II, is most closely shared between Diluvicursor, Anabisetia (MCF-PVPH-74, MCF-PVPH-75; see Fig. S5E–I), Camptosaurus, NMV P18599/NMV P186047 and NMV P186047 (Figs. 19, 20, 31; Figs. S6A–S6D; Table S3). With the exception of Camptosaurus, the distal condyle on mt I in these aforementioned ornithopods is positioned plantar to mt II. The T-shaped distal condyle on mt I of Diluvicursor (Figs. 24A–24F), where the head of the T faces medially, closely resembles the distal condyles in NMV P185992/P185993 and NMV P186047 (Figs. 31A and 31B). This form of condyle is likely possessed by Anabisetia (MCF-PVPH-74), as the displaced bone fragment glued onto the proximal end of mt II (originally thought to be the proximal end of mt I, following Cambiaso, 2007, p. 253, fig. 120) closely resembles the T-shaped distal portion of mt I in Diluvicursor (Figs. 24A–24F; Figs. S5E–S5H). Differing from Diluvicursor, the distal condyles on mt I in Changchunsaurus parvus (Butler et al., 2011), Hypsilophodon (following Galton, 1974, fig. 57J), Othnielosaurus (ROM 46240) and Parksosaurus (ROM 804) are sub-triangular in distal view.

Figure 31 Pedes of selected Eumeralla Formation ornithopods in plantar view.

(A) CT model of right pedal digit I of the Diluvicursor pickeringi gen. et sp. nov. holotype (NMV P221080). (B) Right partial pes of NMV P185992/NMV P185993, NH4Cl coated. (C) Left partial pes of NMV P186047. Dashed arrows in A and B indicate lateral flaring on the cotyle. Abbreviations: ex, extensor groove; fl, flexor groove; mdt, medial distal tarsal; mt #, metatarsal position; pd #, pedal digit number and phalanx position; tib, tibia. Scale bars equal 10 mm.

The proximodistal axis of the phalanges on the right pedal hallux of the D. pickeringi holotype is preserved orthogonal to the long axis of the metatarsal (Fig. 19). However, correct alignment of mt I and pd I-1 on the right pes of NMV P185992/P185993 (Fig. 31B), reveals misalignment of these bones in the D. pickeringi holotype (see restoration, Fig. 31A), as well as on the left pes of NMV P186047 (Fig. 31C). The asymmetrical shape of pd I-1 in D. pickeringi and its dorsoplantar compression are a combination of features uniquely shared with NMV P185992/P185993 (Figs. 24, 31A and 31B), and not evident on pd I-1 of NMV P186047. The asymmetrical form of pd I-1 may have allowed the ungual (pd I-1) to clear the plantomedial edge of mt II.

In D. pickeringi, the width of the diaphysis on mt II is ∼40% that of mt III (Fig. 25). The transversely compressed, lunate (medially convex/laterally concave) diaphysis on mt II of Diluvicursor resembles the condition in Anabisetia (MCF-PVPH-74, Cambiaso, 2007, fig. 120B; Fig. S5E), Gasparinisaura (MUCPv-214, Salgado, Coria & Heredia, 1997, fig. 5.6; MCS-3, M. C. Herne, 2008, personal observation), Morrosaurus (Cambiaso, 2007; Novas, 2009, p. 352; Rozadilla et al., 2016, fig. 5A), NMV P186047 and the dryosaurids Dryosaurus (YPM 1884), Dysalotosaurus (MB.R. 1398), Eousdryosaurus (Escaso et al., 2014), Kangnasaurus and Valdosaurus (following Barrett, 2016, fig. 9D, E) (Fig. 32). In all of these taxa, the transverse width of the diaphysis in the plantar portion of mt II is <50% that of the equivalent region on mt III (Fig. 32). In ornithopods such as Changchunsaurus (Butler et al., 2011), Hypsilophodon (Galton, 1974), Mantellisaurus (NHMUK R11521) and Cumnoria prestwichii (Galton & Powell, 1980; see also McDonald, 2011), the diaphyses on their second metatarsals are transversely compressed (i.e. dorsoplantar heights are greater than transverse widths). However, viewed proximally, the widths in the plantar halves of the diaphyses on mt II of these aforementioned taxa are >60% of the widths on mt III (Fig. 32). The transverse widths of the diaphyses on the second metatarsals of Muttaburrasaurus, Parksosaurus, Talenkauen (Cambiaso, 2007), Tenontosaurus (Forster, 1990) and Thescelosaurus assiniboiensis (Brown, Boyd & Russell, 2011) are sub-equal to mt III, or greater in width, and comparatively blocky (see Fig. 32). In proximal view, the lunate, roughly keyhole-shaped profile of mt II in Diluvicursor most resembles the profiles in Anabisetia and Gasparinisaura, and a similar shape is present in Eousdryosaurus (Figs. 23 and 32). However, transverse compression of mt II in the latter dryosaurid is less than that of the three aforementioned taxa.

Figure 32 Right metatarsi of selected ornithopods in proximal view.

A–R: (A) Diluvicursor pickeringi; (B) NMV P186047; (C) Morrosaurus antarcticus; (D) Gasparinisaura cincosaltensis; (E) Anabisetia saldiviai; (F) Kangnasaurus coetzeei; (G) Eousdryosaurus nanohallucis; (H) Dysalotosaurus lettowvorbecki; (I) Dryosaurus altus; (J) Changchunsaurus parvus; (K) Mantellisaurus atherfieldensis; (L) Hypsilophodon foxii; (M) Tenontosaurus tilletti; (N) Cumnoria prestwichii; (O) Muttaburrasaurus langdoni; (P) Talenkauen santacrucensis; (Q) Parksosaurus warreni; and (R) Thescelosaurus assiniboiensis. Metatarsi normalised for dorsoplantar depth of metatarsal II (shaded black). Dashed lines indicate uncertain bone margins. ?, indicates location of uncertain/expected/missing metatarsal. Abbreviation: mt #, metatarsal position. For data sources, see Table S1.

Ornithischia indet.

Figures S3, 33

Figure 33 Anterior caudal vertebra (NMV P228342) of an indeterminate ornithischian from the ETRW sandstone.

A–F: specimen NH4Cl coated in: (A) left lateral; (B) posterior; (C) right lateral; (D) dorsal; (E) anterior; and (F) ventral views. G–H, schematics in: (G) right lateral; and (H) dorsal views. Vertebral proportions (see also comparisons Fig. 28): ‘a,’ distance from the dorsal tip of the spinal process to the centre of the transverse process base (i.e. neural arch height); and ‘b,’ vertebral height without haemal arch. Abbreviations: acdl, anterior centrodiapophseal lamina; cen, centrum; cprl, centroprezygapophyseal lamina; dap, diapophysis; lcf, laterocentral fossa; sprl, spinopostzygapophyseal lamina; nc, neural canal; ncs, neurocentral suture; pcdl, posterior centrodiapophyseal lamina; podl(p), postzygodiapophyseal lamina (and protuberance); poz, postzygapophysis; pprl, postzygoprezygapophyseal lamina; prdl, prezygodiapophyseal lamina; prsl(p), prespinal lamina (and protuberance); prz, prezygapophysis; sdf, spinodiapophyseal fossa; sp, spinal process; sprf, spinoprezygapophyseal fossa; sprl, spinoprezygapophyseal lamina; spol, spinopostzygapophyseal lamina; spof, spinopostzygapophyseal fossa; tp, transverse process; tprl, transprezygapophyseal lamina; vb, vertebral body. Scale bar equals 50 mm.

Distribution: Lower Cretaceous Australia.

Material: NMV P228342: almost complete isolated caudal vertebra lacking caudal ribs.

Locality: Eric the Red West, ETRW Sandstone, lower Albian, Eumeralla Formation, Otway Group, southern Victoria.

Description

Preservation

NMV P228342 is prepared out, missing the distal-most tip of the spinal process and the caudal ribs (Fig. S3). The spinal process is bent to the left towards its distal end. The distal ends of the transverse processes are eroded or broken and the anterior and posterior margins of the centrum are slightly eroded (Fig. S3).

Morphology

The centrum is amphiplatyan, the anterior and posterior faces round in profile and the laterocentral fossa is shallow. The centrum lacks anterior and posterior haemal facets and the haemal groove is undeveloped. The neurocentral suture is fused and the transverse processes are located on the neural arch. The prezygapophyses are anterodorsally oriented and project only a short distance beyond the centrum. The spinal process is shallowly inclined at 32° from the dorsal plane. The process expands towards its distal end and has a proximodistal length approximately equaling centrum length. The dorsoventral height of the neural arch is ∼50% that of total vertebral height (‘a/b;’ Figs. 28 and 33G). Elliptically shaped postzygapophyses protrude posteriorly from the base of the spinal process. A thin, tab-like prespinal lamina (prsl) is developed anteriorly at the base of spinal process. On the right side, the spinoprezygapophyseal lamina (sprl) connects the prezygapophysis and the base of the spinal process. However, on the left side, the sprl merges with the postdiapophyseal lamina (podl) to form a prezygopostzygapophyseal lamina (pprl), which fails to contact to the spinal process. The podl/pprl forms a thin crista that connects the dorsal margin of the postzygapophysis and the anterior margin of the transverse process, and constitutes the lateral margin of the spinodiapophyseal fossa (sdf) (Figs. 33A, 33C and 33D). A small dorsally oriented protuberance is developed on the podl/pprl, lateral to the sdf. The transprezygapophyseal lamina (tprl) extends between the left and right sprl (Figs. 33D and 33E). The spinopostzygapophyseal lamina (spol) connects the postzygapophysis to the posterior margin of the spinal process (Figs. 33B, 33G and 33H). The paired spols remain separated by a groove-like spinopostzygapophyseal fossa (spof). The prezygodiapophyseal lamina (prdl) connects the prezygapophysis and the dorsal surface of the transverse process (Figs. 33G and 33H) and the centroprezygapophyseal lamina (cprl) extends as a bony sheet from the prdl to the centrum. As a result, the centroprezygapophyseal fossa (cprf) is undeveloped.

Vertebral position

Typically in ornithischians, the spinal processes on the thoracic vertebrae are vertically oriented, anteroposteriorly broad and roughly rectangular in profile (e.g. Heterodontosaurus and Hypsilophodon (Galton, 1974, fig. 22B; Santa Luca, 1980, fig. 5B)). On the thoracic vertebrae of ornithopods, a broadly striated margin developed at both the anterior and posterior ends of the centrum, border the centrolateral fossa (e.g. Dryosaurus (Galton, 1981), Jeholosaurus (Han et al., 2012) and Thescelosaurus neglectus (Gilmore, 1915, fig. 4)). This margin, however, is not typically developed on the caudal vertebrae. The highly reclined spinal process and the lack of striated anterior and posterior margins on the centrum identify NMV P228342 as a caudal vertebra. The lack of facets for haemal arches suggests a caudal position of Ca 1. However, a position of Ca 2 is also possible (e.g. J. shangyuanensis, Han et al., 2012).

Comparisons

Steeply reclined spinal processes of short proximodistal length (approximately equaling centrum length) and a thin, tab-like prsl are features shared between NMV P228342 and the anterior-most caudal vertebrae of D. pickeringi (Figs. 9–11 and 33). The neural arch in NMV P228342 is dorsoventrally higher relative to centrum height than at Ca 3 on the D. pickeringi holotype (i.e. distance ‘a’ relative to distance ‘b;’ Figs. 9A and 33G). However, neural arch heights in NMV P228342 and the anterior-most caudal vertebrae of D. pickeringi are lower than in other ornithopods (Fig. 28). The crista-like podl/pprl on NMV P228342, with its dorsally protrusive process (Fig. 33), may be unique for an ornithischian and possibly for a dinosaur. However, these features could be developed in D. pickeringi, but are unclear from CT imagery of the holotype (Figs. 9B, 10A and 10B). Where a crista-like podl/pprl is developed in NMV P228342, a shallow ridge or bulge is formed in other ornithopods (e.g. Hypsilophodon (Hulke, 1882; Galton, 1974), and Ouranosaurus (Taquet, 1976)), if formed at all. The transversely round profile of the centrum in NMV P228342 (Fig. 33) contrasts with the transversely narrower, elliptical profiles of the anterior-most caudal centra preserved in the D. pickeringi holotype (i.e. Ca 3–4, Figs. 8–10). However, the difference in centrum shape between NMV P228342 and the anterior-most vertebrae in the Diluvicursor holotype could signify different positions in the vertebral series, rather than taxonomic variation (e.g. Hypsilophodon (Galton, 1974, figs. 29B, 31B) and Jeholosaurus (Han et al., 2012, fig. 6A)).

The neural arch on NMV P228342 differs from those on the anterior-most caudal vertebrae of D. pickeringi in several aspects. The anterior and posterior margins of the spinal processes on NMV P228342 expand distally, whereas those of D. pickeringi are parallel. However, distal expansion is only marginal. The prezygapophyses on NMV P228342 are dorsally elevated and anterodorsally directed (Fig. 33), whereas those of D. pickeringi are horizontal, attach at the base of neural arch and positioned lateral to the neural canal (Figs. 9 and 10). NMV P228342 lacks the tuberous process on the sprl of the D. pickeringi neural arches. NMV P228342 likely represents a taxon closely related to D. pickeringi; however, clear morphological differences between the prezygapophyses and sprls of these two ornithischians support their taxonomic separation.

Discussion

Diluvicursor pickeringi nov. gen. et sp., a new small-bodied ornithopod from the locality of Eric the Red West, near Cape Otway, in the lower Albian of the Eumeralla Formation, southeastern Australia, provides new insight on the anatomical diversity of the small-bodied ornithopods from Australia and globally. The holotype (NMV P221080) consists of an almost complete tail, distal portion of the right crus, the complete right tarsus and partial right pes of a turkey-sized individual. These remains were buried in coarse sediments along with substantially sized tree debris that filled scours formed between sand dunes that were migrating downstream in a deep, broad, high-energy river. This deposit is called the ‘ETRW Sandstone.’ An isolated posterior caudal vertebra (NMV P229456) from the same deposit is additionally referred to Diluvicursor pickeringi and pertains to a larger individual than the holotype. A further isolated caudal vertebra (NMV P228342) from the deposit is identified as ∼Ca 1 of an indeterminate ornithischian, but most likely from an ornithopod, closely related to D. pickeringi.

Unusual characteristics of Diluvicursor pickeringi

D. pickeringi is characterised by 10 potential autapomorphies, among which, dorsoventrally low neural arches and transversely broad caudal ribs on the anterior caudal vertebrae are a visually defining combination of features. Typically in ornithischians (e.g. Heterodontosaurus (Santa Luca, 1980, fig. 7), Hypsilophodon (Galton, 1974, figs. 28, 30), Lesothosaurus (‘Stormbergia dangershoeki,’ Butler, 2005, fig. 9A), Thescelosaurus neglectus (Gilmore, 1915, fig. 6)) and dinosaurs in general, the prezygapophyses on the anterior caudal vertebrae are, to some extent, elevated dorsally on the neural arch and, thus, located dorsally relative to the neural canal. However, on the anterior-most caudal vertebrae of D. pickeringi (i.e. at Ca 3–5 on the holotype), the prezygapophyses attach near the base of the neural arches, laterally to the neural canal (Figs. 9 and 10). This unusual morphology in D. pickeringi appears integral to the dorsoventrally low character of the neural arches. Unusually in D. pickeringi, a protuberance is developed on the spinoprezygapophyseal lamina (sprl) of the anterior caudal vertebrae and, as the prezygapophyses are positioned laterally to the neural canal, the transprezygapophyseal lamina (tprl) extends between the paired sprls, approximately level with the dorsal margin of the prezygapophyses (Figs. 9 and 10). The protuberance on the sprl superficially resembles a zygosphene, as in the zygosphene–zygantrum complex in lepidosauromorphs (Romer, 1956, p. 256; Rieppel & Hagdorn, 1997, p. 125; Benton, 2005, p. 150; Tschopp, 2016). However, a structure resembling a zygantrum is not evident in D. pickeringi, in the region of the postzygapophyses.

On the posterior-most caudal vertebra of D. pickeringi (Ca 35–38), triangular intervertebral processes on the anterior articular faces of the centra incise V-shaped notches on the posterior faces of the adjoining centra (Fig. 17). This feature appears to be unique among dinosaurs, although we are presently uncertain whether or not these features are surficial on the centra or developed more extensively across the articular surfaces. Viewed laterally, the centra and the prezygapophyses on the posterior-most caudal vertebrae of Diluvicursor form a roughly herringbone structure (Fig. 17), which superficially resembles the structure formed by the prezygapophyses and haemal arches in ankylosaurs such as Euoplocephalus tutus (Coombs, 1978a, fig. 7). The interlocking vertebral structure in Diluvicursor could have stiffened the posterior end of the tail as an alternative to the ensheathing ossified tendons present in many other ornithopods (e.g. Hypsilophodon, Galton, 1974). Among ornithopods, haemal grooves excavate the ventral surfaces of the caudal vertebrae to varying degrees. Strongly developed grooves present on the middle caudal vertebrae of Diluvicursor, are at least shared with Gasparinisaura. In contrast the grooves are only weakly developed in J. shangyuanensis (Han et al., 2012). Strongly developed haemal grooves, however, present on the posterior-most caudal vertebrae in D. pickeringi appear to be unique, while noting that in many ornithischians, this caudal vertebral region is either poorly known or poorly described.

The lateral distal tarsal of D. pickeringi is embayed by a sulcus, which allowed direct articulation between the calcaneum and mt IV (Figs. 22E–22H). With the exception of stegosaurs (Galton & Upchurch, 2004), direct articulation between the calcaneum and mt IV is unusual for an ornithischian and unknown in any other ornithopod. The asymmetrical form of pd IV-1 in D. pickeringi, where the proximal cotyle is strongly flared medially (Figs. 26G and 26H), is also unusual for an ornithopod, and possibly among dinosaurs (e.g. Coombs, 1978b, fig. 12).

Differentiation of Diluvicursor pickeringi among the Victorian ornithopods

The three previously named ornithopods from Victoria, Atlascopcosaurus, Leaellynasaura and Qantassaurus (Rich & Rich, 1989; Rich & Vickers-Rich, 1999) are only known from cranial remains (see also Herne, Tait & Salisbury, 2016) and whether or not Diluvicursor is synonymous with any of these taxa can only be determined from future discoveries of associated skeletal remains. The only associated ornithopod fossils from Victoria that can be readily compared with D. pickeringi are those of the two partial postcranial skeletons NMV P185992/P185993 and NMV P186047 from Dinosaur Cove. These two partial postcranial skeletons were previously assigned to L. amicagraphica by Rich & Rich (1989) and Rich, Galton & Vickers-Rich (2010), but more recently regarded as indeterminate ornithopods (see Herne, Tait & Salisbury, 2016, for issues regarding these referals).

The skeletal features shared between D. pickeringi and the two aforementioned Dinosaur Cove postcrania will be discussed within ‘the affinities of Diluvicursor pickeringi with comments on phylogenetic datasets,’ below. However, D. pickeringi clearly differs from NMV P185992/P185993 by having a far shorter tail (i.e. ∼50 caudal vertebrae compared to >71 (Herne, 2009)) and from NMV P186047 by having a more robust pes. Relative to NMV P186047, the metatarsus of D. pickeringi is shorter and transversely broader. The spinal processes on the middle caudal vertebrae of D. pickeringi differ from those of NMV P185992/P185993 in being straight, whereas those of the latter recurve dorsally towards their distal ends (Figs. 29A and 29B). Where the haemal processes on the middle caudal vertebrae of D. pickeringi are symmetrically expanded, those of NMV P185992/P185993 are posteriorly expanded (Figs. 29A and 29B). The haemal processes on the middle caudal vertebrae of NMV P186047 further differ from those of both D. pickeringi and NMV P185992/P185993 in being more posteriorly extended and boot-shaped (Fig. 29H). Although more detailed body-form comparisons between the Eumeralla Formation ornithopods require more complete specimens, caudal and pedal morphologies presently suggest that the two Dinosaur Cove ornithopods NMV P185992/P185993 and NMV P186047 were more gracile proportioned ornithopods than D. pickeringi.

Stratigraphic associations of the Eumeralla Formation ornithopods and the status of Diluvicursor pickeringi

The holotype locality of Atlascopcosaurus near Point Lewis is stratigraphically older than the ETRW Sandstone, hosting Diluvicursor (Figs. 1 and 4). These two localities are separated by a true stratigraphic thickness of ∼180 m (Figs. 1 and 4). Dinosaur Cove, which hosts the holotype of Leaellynasaura, NMV P185992/P185993 and NMV P186047 (see Felton, 1997b; Herne, Tait & Salisbury, 2016), is stratigraphically younger than both the ETRW Sandstone and Point Lewis (Figs. 1 and 4). However, apart from palynological studies, which currently indicate that the Eumeralla Formation fossil vertebrate localities fall within ∼3.5 Ma from the beginning of the Albian (following Korasidis et al., 2016), precise chronostratigraphic data for these localities have yet to be published.

Diluvicursor and the Dinosaur Cove ornithopods Leaellynasaura, NMV P185992/P185993 and NMV P186047 are not currently known to be coeval. However, the stratigraphically older taxon A. loadsi (i.e. from Point Lewis), is also known from Dinosaur Cove (Rich & Rich, 1989), including from the Tunnel Sandstone assemblage (see Herne, Tait & Salisbury, 2016). Thus, as the stratigraphic range of Atlascopcosaurus extends through the ETRW Sandstone, Atlascopcosaurus and Diluvicursor are coeval. However, whether or not Diluvicursor and Atlascopcosaurus are synonymous can only be determined from future fossil discoveries, where anatomical congruence might be demonstrated. Importantly, the presence of the isolated caudal vertebra NMV P228342 (Fig. 33) in the fossil assemblage of the ETRW Sandstone, identified as an indeterminate ornithischian with morphology clearly differing from D. pickeringi, also has a bearing on the status of D. pickeringi. The presence of NMV P228342 indicates that the ETRW Sandstone hosts more small-bodied ornithischians than Diluvicursor and strengthens our view that the locality potentially hosts both Diluvicursor and Atlascopcosaurus, and the inclusion of other ornithopod taxa cannot be discounted.

The affinities of Diluvicursor pickeringi with comments on phylogenetic datasets

The potential phylogenetic relationships of D. pickeringi were analyzed within the data matrices of Boyd (2015), Dieudonné et al. (2016) and Han et al. (2017). No characters were excluded or modified from those presented by the aforementioned authors and no new characters added (scores for D. pickeringi and datasets utilized are provided in Table S4 and Dataset S1). As far as possible, the analyses used the search parameters originally described, including the ordering of 9 and 21 characters in the Dieudonné et al. (2016) and Han et al. (2017) datasets, respectively; however, the search parameters in Boyd (2015) were modified, as the number of replications (10,000) and trees held (10,000) per replication were considered impractical (for further details, see Fig. S7).

An initial search derived from the dataset of Boyd (2015) using all of the originally included operational taxonomic units (OTU), plus D. pickeringi, resulted in a poorly resolved strict consensus tree (Fig. S7A). From this analysis, D. pickeringi occupied a position at the base of Cerapoda in a polytomy with 16 other taxa. A subsequent search further modified the matrix of Boyd (2015), altering the character scores for Atlascopcosaurus and Leaellynasaura to reflect current taxonomic understanding (Herne, 2014; Herne, Tait & Salisbury, 2016; see Fig. S7; Table S4; Dataset S1). In addition, a maxillary tooth character, scored for Q. intrepidus by Boyd (2015), was also removed (see Fig. S7). This search showed that D. pickeringi occupied a polytomous position within Cerapoda along with nearly all other cerapodan OTUs (Fig. S7B) and was more unresolved than the first search. A subsequent search derived from the matrix of Dieudonné et al. (2016) resulted in the position of Diluvicursor in a large clade forming a polytomy within Ornithopoda, as sister clade to Orodromeus. Here, both Ornithopoda and the unnamed clade containing D. pickeringi lacked significant resampling support (Fig. S8). A final search was derived from the matrix of Han et al. (2017), using all of the originally included OTUs, plus D. pickeringi, to yield a strict consensus (Fig. S9A). An additional consensus from this search was produced after eight OTUs were pruned from the strict consensus tree (a posteriori taxon removals, as per Han et al., 2017, fig. 16. Both consensus trees place D. pickeringi within polytomies at the base of Ornithischia (Figs. S9A and S9B).

Despite D. pickeringi exhibiting a high number of unique characters (or combinations thereof), based on limited material, its position in the analyses we conducted was unstable, weakly supported, and unresolved within Cerapoda or Ornithopoda. Generally, inclusion of D. pickeringi resulted in consensus trees with worse structure and weaker clade support than those originally reported. The poor resolution of taxa in these results demonstrates the destabilising effect of highly incomplete OTUs, such as D. pickeringi (see Butler, Upchurch & Norman, 2008; Han et al., 2012). It is further apparent global datasets may be impractical for addressing phylogenetic inquiries regarding new OTUs of interest appended to revised iterations. Typically, with the addition of new OTUs to existing datasets, new characters are added and some existing characters or codings are modified (Han et al., 2017). However, modifying an existing dataset, such as those of Boyd (2015), Dieudonné et al. (2016) and Han et al. (2017), to include new and modified characters/codings, was beyond the scope of the present contribution. A new dataset with special emphasis on the Australian ornithopods is currently being prepared (M. Herne, J. Nair, 2014–2017, unpublished data) and will be published elsewhere.

Pending analysis of the revised dataset, the following features of D. pickeringi are potentially synapomorphic. Centrum length on the anterior-most middle caudal vertebrae of ∼50% that of total dorsoventral vertebral height (Fig. 29), is shared with NMV P185992/P185993. Similar centrum length to vertebral height is evident in Gasparinisaura (MUCPv-212, 42%) and Valdosaurus (IWCMS 2013.175, 37%, following Barrett, 2016, fig. 4), while noting that in the latter taxon the spinal processes are lengthier (Figs. 29C and 29D). Anteroposteriorly expanded haemal processes on the middle caudal vertebrae of Diluvicursor, where the processes expand abruptly from the shaft (Figs. 12–14) rather than flaring gradually towards their distal ends (e.g. Haya, Makovicky et al., 2011), are shared with NMV P185992/P185993 and NMV P186047 (Herne, 2014, fig. 9.24), Gasparinisaura (Coria & Salgado, 1996, fig. 4; MCS-1, M. C. Herne, 2008, personal observation), Parksosaurus and Macrogryphosaurus (Calvo, Porfiri & Novas, 2007 fig. 5). However, the small size of the expanded haemal processes on the middle caudal vertebrae of D. pickeringi more closely resemble those of NMV P185992/P185993 and NMV P186047 than the other taxa mentioned, where the processes are anteroposteriorly broader and proximodistally deeper.

Asymmetrically expanded, boot-shaped haemal processes on the posterior caudal vertebrae of D. pickeringi are shared with Camptosaurus (Gilmore, 1909, fig. 35, plate 17), NMV P185992/P185993 and potentially NMV P186047. Unlike Diluvicursor, asymmetrically expanded haemal processes are known on the middle caudal vertebrae of Gasparinisaura and Macrogryphosaurus, as in NMV P185992/P185993 and NMV P186047. However, in Gasparinisaura and Macrogryphosaurus, haemal processes are presently unknown on the posterior caudal vertebrae and thus, cannot be compared with the Australian taxa. The haemal processes of D. pickeringi present a series of shape changes along the caudal vertebral series. The identification of another ornithischian that shares the same combination of haemal process shapes with D. pickeringi, and in the same vertebral regions, is presently unknown. However, the shapes and proportions of the haemal processes on the caudal vertebrae of D. pickeringi, particularly in the posterior-most middle to posterior caudal regions, are closest to NMV P185992/P185993 and NMV P186047.

A transversely broad, proximally obtuse, centrally positioned anterior ascending process, on the astragalus of Diluvicursor, resembles the processes in Anabisetia, Muttaburrasaurus and Dryosaurus (Fig. S5). However, a fossa bordering the lateral margin of the processes in the latter three taxa (Fig. S5) is not evident in D. pickeringi. A thin, wavy, sub-rectangular medial distal tarsal with a grooved proximal surface may be uniquely shared with NMV P186047 (Figs. 30D–30F).

In D. pickeringi, the transverse and dorsoplantar proportions of pd I-1 and the distal condyle on mt I are relatively reduced (gracile) and within 60% of the sizes of the equivalent regions on pedal digit II (Fig. S6). These proportions on the hallux of Diluvicursor are closer to those in Anabisetia (MCF-PVPH-75), Camptosaurus (Gilmore, 1909, fig. 35, plate 17) and NMV P186047 than the other ornithischians presently assessed (Figs. 19, 20, 24F and 31; Figs S5G, S6A–D; Table S3). Similar proportions of distal mt I and pd I-1 are observed in NMV P185992/P185993 (Fig. S6), while noting that the relative dorsoplantar heights of pedal digits I and II are unknown. The T-shaped distal condyle on mt I of Diluvicursor and its position plantar to mt II, is shared with NMV P185992/P185993, NMV P186047 and provisionally, Anabisetia (see Fig. S5). A similarly formed condyle distally on mt I, could also be present in the early neornithischian L. diagnosticus (following Sereno, 1991), but this possible similarity requires further examination. Asymmetric expansion of pd I-1 is uniquely shared with NMV P185992/P185993 (Fig. 31).

A transversely compressed diaphysis and lunate proximal profile on mt II of Diluvicursor, are features shared with Anabisetia, Gasparinisaura, NMV P186047, Morrosaurus and the dryosaurids Dryosaurus, Dysalotosaurus, Eousdryosaurus, Valdosaurus and potentially Kangnasaurus (Figs. 23C, 25B, 25F and 32; Fig. S5E). The diaphyses on the second metatarsals of Hypsilophodon and Changchunsaurus are also transversely compressed, but differ from those of the aforementioned ornithopods by having diaphyseal widths sub-equal to mt III and they also lack the lunate proximal profile (Fig. 32). In Mantellisaurus, Muttaburrasaurus, Parksosaurus, Talenkauen, Thescelosaurus assiniboiensis and Tenontosaurus, the diaphysis on mt II is transversely broader, relative to mt III, and in some of these taxa, blocky (see Fig. 32). A roughly keyhole shaped profile at the proximal end of mt II (Figs. 23C and 25F) is shared with Anabisetia and Gasparinisaura (Fig. 32; Fig. S5E). The proximal end of mt II in NMV P185992/P185993 is not preserved and cannot be compared.

In summary, analysis of the datasets of Boyd (2015) and Dieudonné et al. (2016) positioned D. pickeringi in either Cerapoda or Ornithopoda, respectively (Figs. S7 and S8). Analysis of the dataset of Han et al. (2017) placed D. pickeringi in a polytomy at the base of Ornithischia (Fig. S9). With the inclusion of D. pickeringi in the original dataset of Boyd (2015), Cerapoda is supported by at least 10 synapomorphies and with scores for the Eumeralla Fm OTUs either updated or corrected, Cerapoda is supported by 21 synapomorphies. With the inclusion of Diluvicursor in the dataset of Dieudonné et al. (2016), Ornithopoda is supported by five synapomorphies and within this clade, a more exclusive unnamed node containing OTUs more nested than Orodromeus is diagnosed by 16 synapomorphies. It is of note, however, that although D. pickeringi is recovered in these consensus positions, none of the synapomorphies for the nodes recovered (see Figs. S7 and S8 for listings) are characters actually scored for D. pickeringi. Consequently, based solely upon these analyses, we currently consider the assignment of D. pickeringi to Cerapoda/Ornithopoda extremely tenuous.

From qualitative observations, a close relationship between D. pickeringi and the taxon or taxa represented by NMV P185992/P185993 and NMV P186047 is suggested. Features of the caudal vertebrae, astragalus and pes (i.e. dorsoventrally low vertebral proportions; a broad obtuse anterior ascending process on the astragalus; a gracile, plantarly positioned hallux, with a T-shaped distal condyle on mt I; and a lunate, transversely compressed mt II), variously possessed by the Eumeralla Formation ornithopods, the South American ornithopods Anabisetia and Gasparinisaura, Afro-Laurasian dryosaurids and possibly the Antarctic ornithopod Morrosaurus, suggest close phylogenetic affinities potentially exist between these taxa. However, future cladistic analysis will help to test these hypothesised relationships.

Pedal pathologies of the Diluvicursor pickeringi holotype

Features on the right pes of the Diluvicursor pickeringi holotype suggest this individual may have endured antemortem injury. As preserved, the proximodistal axis of pd I-1 is deflected medially relative to the proximodistal axis of mt II (Fig. 34A). Disarrangement (angulation and dislocation) of the phalanges and metatarsals can, of course, occur postmortem through taphonomic processes. However, antemortem angulation at the metatarsophalangeal (mtp) joint—a condition termed subluxation resulting from trauma or disease (following Burgener, Kormano & Tomi, 2006)—rather than taphonomic disruption, is supported by the identification of rugosely textured bone, which appears to have formed a flange-like overgrowth on the proximal margin of pd II-1, enveloping the distal condyle on mt II (Fig. 34B). The suggested bone overgrowth on pd I-1, termed osteophytosis (see Resnick, 1983; Burgener, Kormano & Tomi, 2006, p. 166), could have helped to stabilize the mtp joint following trauma (Lieben, 2016). Osteophytosis on pd II-1 may have resulted in limited mobility of the mtp joint. It is possible that the joint had been immobile. Further investigation on pedal pathologies in the D. pickeringi holotype could benefit from non-invasive histological examination, using techniques such as synchrotron radiation X-ray microtomography (Curtin et al., 2012).

Figure 34 Diluvicursor pickeringi gen. et sp. nov. holotype (NMV P221080), potential pathologies of the right pes.

(A) CT model of the pes in plantolateral view, with pedal digits I and V removed. (B) Metatarsophalangeal joint on pedal digit II in plantolateral view. Dashed lines in A indicate deflected axes between mt IV and pd IV-1. Dotted line in B indicates rugose bone on the proximal margin of pd IV-1. Dashed arrows in A–B indicate areas of osteophytosis (also see photographs, Figs. 19A and 19B). Abbreviations: mt #, metatarsal position; pd #, pedal digit number and phalanx position. Scale bars equal 10 mm.

Ontogeny and body-size of Diluvicursor pickeringi

Restoration of the Diluvicursor pickeringi holotype (Fig. 7) suggests that the total anteroposterior length of this individual was ∼1.2 m. Unfused anterior caudal vertebrae on the holotype further suggest this individual was a juvenile (Hone, Farke & Wedel, 2016). However, osteophytosis of the right pes also suggests that the holotype was of sufficient age to have recovered from traumatic subluxation of the pedal digits. The size of the isolated posterior caudal vertebra NMV P229456, referred to D. pickeringi (Fig. 21), further suggests that the taxon grew to at least 2.3 m in length. However, whether or not NMV P229456 pertains to an adult is unknown.

Anterior caudal myology of Diluvicursor pickeringi

In the anterior caudal region of non-avian dinosaurs, the epaxial and hypaxial musculature are located dorsally and ventrally to the caudal ribs, respectively (Fig. 35) (Mallison, Pittman & Schwarz, 2015). The epaxial musculature likely comprised the musculus (M.) dorsalis caudae (see Galton, 1974; Norman, 1986; Mallison, Pittman & Schwarz, 2015; in crocodilians, the M. transversospinalis and M. longisimus caudalae/dorsi, following Organ, 2003; Persons & Currie, 2014; see also Persons, Currie & Norell, 2014), while the hypaxial musculature likely comprised the M. rectus abdominus, M. ilio-ischiocaudalis, M. transversus perinei and M. caudofemoralis longus, with the latter muscle integral to locomotory function of the hind limb (Maidment & Barrett, 2011; Maidment et al., 2014; Persons & Currie, 2014; Persons, Currie & Norell, 2014; Mallison, Pittman & Schwarz, 2015). The caudal ribs on the anterior-most caudal vertebrae of D. pickeringi (e.g. Ca 3; Fig. 35) indicate that the musculature in this region was transversely broad. The dorsoventrally low neural arches in this region of the tail, suggests that the epaxial musculature was dorsoventrally shallow, while the hypaxial musculature was dorsoventrally deep.

Figure 35 Schematic transverse section through the anterior epaxial and hypaxial muscular regions of the ornithopod tail.

(A) Diluvicursor pickeringi gen. et sp. nov. holotype (NMV P221080), designated Ca 3 (see also Fig. 9); and (B) Hypsilophodon foxii, Ca 4 (NHMUK R196; following Galton, 1974, figs. 28–29). Sections normalised for total vertebral depth. Abbreviations: Ca #, caudal vertebra and position. Scale bars equal 10 mm.

Differing from the proportionately deep hypaxial locomotory musculature in the tail of D. pickeringi, the dorsoventral heights of the epaxial and hypaxial musculature in the tail in Hypsilophodon were likely to have been sub-equal (e.g. Ca 4, based on Galton (1974); Fig. 35). Furthermore, in comparison to D. pickeringi, the width across the caudal ribs in H. foxii suggests that the musculature in this region was transversely narrower than that of D. pickeringi (Fig. 35). The differences between the anterior caudal musculature of Diluvicursor and Hypsilophodon could signify differing locomotor abilities between these two taxa, and highlight an area for future comparative investigations of ornithopod locomotion.

It is interesting to note that the proportions of the epaxial and hypaxial musculature in the anterior caudal region of Diluvicursor resemble those in the oviraptorosaur Heyuannia yanshini (Funston et al., in press). In H. yanshini, neural arch height is ∼22% of the total vertebral height and the transverse width across the caudal ribs, ∼75% of total vertebral height (following Persons, Currie & Norell, 2014). These vertebral proportions in H. yanshini have been considered unusual in a theropod (Persons, Currie & Norell, 2014). Calculation of relative femoral adductor muscle mass (M. caudofemoralis longus) in H. yanshini, against body weight, suggested a taxon with substantial running ability (sensu Persons, Currie & Norell, 2014). Although we cannot estimate the body mass of the Diluvicursor holotype (e.g. the femur, from which body-mass has typically been calculated (Anderson, Hall-Martin & Russell, 1985; Campione et al., 2014), is unknown), similarity in the proportions of the caudal hypaxial musculature between Diluvicursor and Heyuannia suggests these two taxa could have shared similarly strong locomotory abilities.

Palaeoecological context of Diluvicursor pickeringi

A rich assemblage of isolated vertebrate fossils has been reported from Eric the Red West, including those of fishes, chelonians, plesiosaurs, pterosaurs, small ornithischians, theropods and mammals (Rich, 2015) (Fig. 36). However, apart from the new ornithischians described in this present work, an indeterminate spinosaurid cervical vertebra (NMV P221081, Barrett et al., 2011a; Fig. 5B) and a mammalian mandible fragment (NMV P228848) referred to the ausktribosphenid, cf. Bishops whitmorei (Rich et al., 2009b), much of the fossil material from this locality has yet to be published. The description of D. pickeringi significantly adds to the growing body of information on the tetrapods from this site and region. Importantly, the ETRW Sandstone sheds new light on the palaeoecosystem of the Australian-Antarctic rift graben, within which D. pickeringi and other biota coexisted.

Figure 36 Artist’s interpretation of the early Albian, volcaniclastic, floodplain palaeoenvironment within the Australian-Antarctic rift graben, in the region of Eric the Red West.

Scene depicting two individuals of Diluvicursor pickeringi on the cutbank of a high-energy meandering river, regional floral components and distant rift margin uplands. Floral components potentially included forest trees of Araucariaceae (Agathis and Araucaria), Podocarpaceae and Cupressaceae and lower story/ground cover plants, including pteridophytes (ferns, including equisetaleans), hepatics, lycopods, cycadophytes, bennettitaleans, seed-bearing fern- or cycad-like taeniopterids and early Australian angiosperms. Artwork by P. Trusler, with permission.

Scours at the base of the large river that the ETRW Sandstone represents filled with coarse bedload containing mudstone rip-up clasts, medium and coarse sand, quartzose gravel/grit and sizable, compressed, coalified/carbonized woody plant fossils, including river transported logs and tree stumps (see also Rich et al., 2009b) (Figs 3, 5). The fossil plant materials in these sediments suggest that the river incised a forested floodplain (Fig. 36). Previous macrofloral and palynological investigations indicate that conifers, principally Araucariaceae (Agathis and Araucaria), Podocarpaceae and Cupressaceae, were the dominant forest tree types in the Eumeralla Formation during the late Aptian–early Albian (Douglas, 1969; Douglas & Williams, 1982; Wagstaff & McEwan Mason, 1989; Dettmann et al., 1992; Dettmann, 1994; Korasidis et al., 2016). The logs and tree stumps potentially pertain to these tree types. The sizes of some of the logs in ETRW Sandstone (one almost 1 m in diameter) further suggest old-growth forests were established in the Eumeralla Formation, with trees that were potentially several hundreds of years in age (see also Seegets-Villiers, 2012, on deposits in the Wonthaggi Fm).

A complex assemblage of lower story plants (understory, groundcover and shallow-water aquatic plants) was also present in the Eumeralla Formation during the Albian, including terrestrial and aquatic pteridophytes, hepatics, lycopods, cycadophytes, bennettitaleans, seed-bearing fern- or cycad-like taeniopterids and non-magnoliid dicotyledonous angiosperms (Douglas, 1969, 1973; Douglas & Williams, 1982; Dettmann et al., 1992; Dettmann, 1994). Palynological investigations further show that early Australian angiosperms became increasingly more complex at stratigraphically younger localities within the Eumeralla Formation during the Albian (Korasidis et al., 2016). From these data combined, at the time of D. pickeringi and in the region of Eric the Red West, well-established conifer forests and complex lower story plant assemblages were likely.

The Lower Cretaceous forests on the Australian–Antarctic rift floodplain would have been interspersed by large, deep rivers with broad inner banks and shallow floodplain lakes (see Fig. 36). These hydrological features, evident from the ETRW and Anchor sandstones, likely supported varied vegetation zones with complex faunal habitat opportunities. The migrating banks of the meandering rivers would have provided ideal conditions for vegetation successions, as in modern systems (Hickin, 1974), and periodic disturbance of the forests by overbank flooding would have created local physiographic differences. Similarly to present-day floodplain ecosystems (Junk, Bayley & Sparks, 1989; Baker & Barnes, 1998; Whited et al., 2007; Tockner et al., 2008), a mosaic of vegetation zones likely characterised the Early Cretaceous floodplain in the region of ETRW. We speculate that periodic disturbance of older forests through flooding and the migration of high-energy rivers, such as that represented by the ETRW Sandstone, potentially favoured opportunistic pteridophytes, cycadophytes and angiosperms (Fig. 36). The dynamics of change in physiography and vegetation on the rift floodplain in the region of ETRW would have provided varied niche opportunities for dinosaurian herbivores, such as D. pickeringi, and predators alike.

Past investigations on co-occurring ornithischian herbivores, particularly large-bodied ankylosaurs, ceratopsians and hadrosaurs have aimed to identify ecomorphological reasons underlying disparity in dental and cranial features between taxa, both within and between clades. (Fricke & Pearson, 2008; Henderson, 2010; Mallon & Anderson, 2013). Thus, differing dental morphologies, cranial structures and jaw biomechanics potentially signified differing niche selection preferences between co-occurring taxa. However, the ecomorphological implications of cranial and dental disparity between co-occurring small-bodied ornithopods, let alonepostcranial disparity, are areas of research that have yet to be investigated. Morphological disparity between the postcranial skeleton of D. pickeringi and those of the stratigraphically younger ornithopods from Dinosaur Cove (NMV P185992/P185993 and NMV P186047) could hold palaeoecological significance. D. pickeringi appears to have been a more robust ornithopod than the Dinosaur Cove ornithopods, represented by articulated postcranial remains (i.e., NMV P185992/P185993 and NMV P186047). In addition, the sizes of the palaeorivers within which the individuals to which these postcrania pertain, were buried and preserved, also differ significantly. The river in which the Dinosaur Cove ornithopods were deposited (see Herne, Tait & Salisbury, 2016) was substantially smaller and of lower hydraulic power than that in which D. pickeringi was buried and the ETRW Sandstone represents. The differences between these deposits could imply differing palaeoecological conditions. The morphological differences between the Eumeralla Formation ornithopods could signify differing niche selection preferences. These ornithopods therefore provide significant materials for future research on the ecomorphology of small-bodied, non-avian dinosaurs from Gondwana.

Conclusion

Diluvicursor pickeringi nov. gen. et sp. is a new small-bodied ornithopod from the lower Albian of the Eumeralla Formation in the Otway Basin. The taxon is known from an almost complete tail and lower partial right limb of the holotype (NMV P221080), as well as an isolated posterior caudal vertebra (NMV P229456), discovered at the fossil locality of Eric the Red West (ETRW). The deposit, termed the ETRW Sandstone, is interpreted to have been a broad (∼600 m), deep (∼25 m), high-energy meandering river. Sediments and fossils from the ETRW Sandstone indicate that D. pickeringi inhabited a faunally rich, substantially forested riverine floodplain within the Australian–Antarctic rift complex. A further isolated caudal vertebra from the deposit (NMV P228342), interpreted as that of an indeterminate ornithischian, suggests the locality may have hosted at least two small-bodied ornithischians. D. pickeringi grew to at least 2.3 m in length and is characterised by 10 potential autapomorphies, among which, the combination of dorsoventrally low neural arches and transversely broad caudal ribs on the anterior-most caudal vertebrae present a visually defining combination of features.

Features of the caudal vertebrae and pes suggest that D. pickeringi and the two stratigraphically younger, indeterminate ornithopods from Dinosaur Cove, NMV P185992/P185993 and NMV P186047, are closely related. However, D. pickeringi differs from NMV P185992/P185993 by having a far shorter tail (50 vertebrae compared to >71) and from NMV P186047 by having a comparatively shorter, more robust, pes. The phylogenetic position of D. pickeringi investigated through searches within three recently published datasets was unresolved beyond placement within a polytomous clade of non-iguanodontian ornithopods. Various features of the caudal vertebrae and pes suggest that the Eumeralla Formation ornithopods Diluvicursor, NMV P185992/P185993 and NMV P186047 may be more closely related to the Argentinean ornithopods Anabisetia and Gasparinisaura, the Antarctic ornithopod Morrosaurus and possibly Afro-Laurasian dryosaurids, than all other ornithopods. A common progenitor of these taxa is suggested. However, these suggested affinities are to be tested more rigorously within a revised cladistic dataset of Gondwanan ornithopods.

The discovery of D. pickeringi in the ETRW Sandstone indicates that future prospecting efforts in the Eumeralla Formation at locations where coarse, gritty sediments crop-out at the base of deep palaeoriver channels, could lead to significant new discoveries (see also Rich et al., 2009b). The articulated postcrania of similarly sized, but anatomically differing small-bodied ornithopods from the Eumeralla Formation provide unique fossil material for future comparative investigations on dinosaur biomechanics, and how differing locomotor abilities could relate to differing palaeoecosystems.

Supplemental Information

Supplemental Information 1 Map of East Gondwana at ∼113 Ma in the region of Australia and Antarctica.

Click here for additional data file.

Supplemental Information 2 Relative ages of Eumeralla Formation fossil vertebrate localities.

Click here for additional data file.

Supplemental Information 3 Taphonomic features of two isolated caudal vertebrae from the ETRW Sandstone.

(A) NMV P228342 in right dorsolateral view. (B) NMV P229456 in left lateroventral view. Abbreviations: cen, centrum; poz, postzygapophysis; prz, prezygapophysis; sp, spinal process; tp, transverse process. Scale bar 1 cm.

Click here for additional data file.

Supplemental Information 4 Sediments of the ETRW Sandstone showing matrix supported conglomerate hosting the partial postcranium NMV P221080.

(A) Eroded vertical surface on block ‘B1’ looking north in the region of the anterior caudal vertebrae. (B) Top view of block ‘B1’ in the region of the right pes. (C) Top view of block ‘B5’ in the region of the posterior caudal vertebrae. Abbreviations: Ca #, designated caudal vertebra and position; cpd, coalified plant debris; g, gravel/grit; mc, mudrock clast. Scale bar in A, 1 cm. Scale increments in B–C, 1 cm.

Click here for additional data file.

Supplemental Information 5 Lower hind limb skeletal features in selected ornithopods.

A–B, distal left crus and proximal tarsus of the Muttaburrasaurus langdoni holotype (QM F6140) in anterior view: (A) image; and (B) schematic. (C) Distal left crus and proximal tarsus of D. altus (YPM 1876, cast) in anterior view. D–G, Anabisetia (MCF-PVPH-74): (D) distal left crus and proximal tarsus in anterior view; (E) left metatarsus in proximal view; (F) proximal region of left metatarsus in medial view; and (G) schematic of postulated distal mt I in plantomedial view (shown separated from the proximal end of mt II). (H) CT model of distal left mt I of the D. pickeringi holotype (NMV P221080) in in plantomedial view. (I) Right pes of Anabisetia (MCF-PVPH-75) in medial view. Abbreviations: aad, adductor surface; aap, anterior ascending process of astragalus; as, astragalus; cal, calcaneum; ex, extensor groove; fib, fibula; fos, fossa; mt #, metatarsal and position; pd #, pedal phalanx, number and phalanx position; tib, tibia. Scale bars: A and G, 10 cm; B, 5 cm; C–F, 1 cm.

Click here for additional data file.

Supplemental Information 6 Pedal digit proportions for selected ornithopods and early ornithischians.

(A) Distal dorsoplantar height of metatarsal I, relative to metatarsal II. (B) Distal transverse width of mt I relative to mt II. (C) Proximal dorsoplantar height of pd I-1 relative to pd II-1. (D) Proximal transverse width pd I-1 relative to pd II-1. Abbreviations: e, proximal pd II-1 estimated from distal mt I; mt #, metatarsal position; pd #, pedal phalanx position. Data sources, Table S1; tabulated data, Table S3.

Click here for additional data file.

Supplemental Information 7 D. pickeringi in strict consensus trees derived from the matrix of Boyd (2015).

Click here for additional data file.

Supplemental Information 8 D. pickeringi in strict consensus tree derived from the matrix of Dieudonné et al. (2016).

Click here for additional data file.

Supplemental Information 9 D. pickeringi in consensus trees derived from the matrix of Han et al. (2017).

Click here for additional data file.

Supplemental Information 10 Fossil taxa/materials examined or compared in this work with information on occurrence, primary literature sources and additional image resources utilized.

Abbreviations: ETRW, Eric the Red West.

Click here for additional data file.

Supplemental Information 11 Comparative dorsoventral proportions of the anterior caudal vertebrae for selected ornithopods (see Fig. 28).

Dorsoventral heights: ‘a’ measured vertically from dorsal tip of spinal process to centre of transverse process; ‘b,’ measured vertically from dorsal tip of spinal process to ventral-most margin of centrum; and ‘c,’ measured from dorsal tip of spinal process to ventral tip of haemal process. For taxa where height information is not shown, proportions were estimated from figures within the literature (see sources, Table 1). Abbreviations: Ca, caudal vertebral position; ?, unknown or estimated caudal position.

Click here for additional data file.

Supplemental Information 12 Pedal digit proportions for selected ornithischians (see Figs S6–S7).

Notes: the first pedal phalanges (pd I-1) in D. lettowvorbecki and Eousdryosaurus nanohallucis are alternative identifications using the dimensions for the bones identified as first metatarsals (following Galton, 1981; Escaso et al., 2014). Measurements of elements from literature sources in Table S1. Abbreviations: DDH, distal dorsoplantar height; DTW, distal transverse width; (e), estimated from articulating surface of adjoining bone; l, left; mt #, metatarsal position; pd #, phalanx and position; PDL, proximodistal length; PDH, proximal dorsoplantar height; PTW, proximal transverse width; r, right.

Click here for additional data file.

Supplemental Information 13 Character states of new OTUs added to the matrices published by Boyd (2015), Dieudonné et al. (2016) , and Han et al. (2017, in press).

Click here for additional data file.

Supplemental Information 14 Ornithischian taxon/character datasets.

Click here for additional data file.

The authors acknowledge the Eastern Maar peoples as the traditional custodians of the research area, and pay respects to their elders past and present. We thank George Caspar for the discovery of the Diluvicursor pickeringi holotype and the field volunteers of Dinosaur Dreaming who participated in the excavations at Eric the Red West. We gratefully thank T. Rich for making NMV P221080 available for research, L. Kool (MU) and D. Pickering (MV) for preparation of the fossils in this work, and the Geological Sciences staff of Museums Victoria for their assistance and resources that helped this work to proceed. For access to specimens in their care, we thank R. Coria (MCF-PVPH), J. Calvo, J. Porfiri and R. Juérez Valieri (Centro Paleontológico Lago Barreales), I. Cerda (MCS), L. Salgado (MUCPv), P. Barrett, S. Chapman (NHMUK), K. Spring (QM), K. Seymour and D. Evans (ROM), and R. Martinez (UNPSJB). We are extremely grateful to S. O’Hara and St. Vincent’s Hospital, Melbourne for the CT scanning of NMV P221080. The authors thank P. Trusler for production the painting in Figure 36. We thank T. Rich, P. Vickers-Rich, and W. White for editing advice that helped improve this paper, and A. Abell, S. Bryan, P. Chedgey, C. Coronel, S. Edwards, D. Evans, F.-L. Han, D. Henry, D. Herne, V. Korasidis, M. Lamanna, M. Lyon, A. Maguire, F. Novas, S. Poropat, J. Rosine, E. Tschopp, D. Seegets-Villiers, M. Walters, D. Schwarz, W. White and T. Ziegler for assistance, discussion, additional resources, advice or support that helped make this work possible. The authors gratefully thank A. Farke (Academic Editor) and Reviewers C. Brown, K. Poole and an anonymous reviewer, whose comments and suggestions greatly improved this manuscript. MCH thanks P. Currie and L. Salgado for suggestions stemming from examination of his unpublished PhD thesis that preceded this work.

Abbreviations

Anatomical/technical

Ca # caudal vertebra and designated/estimated position

CT computed tomography

M. musculus

mpt metatarsophalangeal

mt # metatarsal number or range

NH4Cl ammonium chloride

pd # pedal digit number and phalanx position

Abbreviations for vertebral laminae and fossae are provided in Table 1.

Institutional

CMN Canadian Museum of Nature, Ottawa, Ontario, Canada

DD Volunteers, Monash University and Museums Victoria staff of the Dinosaur Dreaming project, Victoria, Australia

IWCMS Dinosaur Isle, Sandown, United Kingdom

MACN Coleccion Paleontología de Vertebrados, Museo Argentino de Ciencias Naturales ‘Bernardino Rivadavia,’ Buenos Aires, Argentina

JLUM Geological Museum of the Jilin University, Changchun, Peoples Republic of China

MB.R. Collection of Fossil Reptilia, Museum für Naturkunde (MfN), Berlin, Germany

MCF-PVPH Museo Carmen Funes-Paleontología de Vertebrados, Plaza Huincul, Neuquén Province, Argentina

MCS Museo Cinco Saltos

MU Monash University, Melbourne, Victoria, Australia

MPM Museo Padre Molina, Rio Gallegos, Santa Cruz, Argentina

MUCPv Museo de Geologia y Paleontologia de la Universidad Nacional del Comahue, Paleontologia de Vertebrados, Neuquén Province, Argentina

MV Museums Victoria, Melbourne, Victoria, Australia (formerly, National Museum of Victoria (NMV))

NHMUK Natural History Museum, London, United Kingdom (formerly the British Museum of Natural History)

OUMUK Oxford University Museum of Natural History, Oxford, United Kingdom

QM Queensland Museum, Brisbane, Queensland, Australia

ROM Royal Ontario Museum, Toronto, Ontario, Canada

RSM Royal Saskatchewan Museum, Regina, Canada

SAM South African Museum, Cape Town, South Africa

SHN Sociedade de História Natural, Torres Vedras, Portugal

UNPSJB Universidad Nacional de la Patagonia ‘San Juan Bosco,’ Argentina

USNM National Museum of Natural History, Washington, DC, United States

YPM Yale Peabody Museum, New Haven, Connecticut, United States

Geographical/Geological

az azimuth

ETRW ‘Eric the Red West’

Fm formation

Ma mega annum (millions of years)

Additional Information and Declarations

Competing Interests

Author Contributions

Data Availability

New Species Registration

The authors declare that they have no competing interests.

Matthew C. Herne conceived and designed the experiments, performed the experiments, analyzed the data, contributed reagents/materials/analysis tools, wrote the paper, prepared figures and/or tables, reviewed drafts of the paper.

Alan M. Tait conceived and designed the experiments, performed the experiments, analyzed the data, contributed reagents/materials/analysis tools, wrote the paper, prepared figures and/or tables, reviewed drafts of the paper.

Vera Weisbecker conceived and designed the experiments, performed the experiments, analyzed the data, contributed reagents/materials/analysis tools, wrote the paper, prepared figures and/or tables, reviewed drafts of the paper.

Michael Hall conceived and designed the experiments, performed the experiments, analyzed the data, contributed reagents/materials/analysis tools, wrote the paper, prepared figures and/or tables, reviewed drafts of the paper.

Jay P. Nair conceived and designed the experiments, performed the experiments, analyzed the data, contributed reagents/materials/analysis tools, wrote the paper, prepared figures and/or tables, reviewed drafts of the paper.

Michael Cleeland performed the experiments, contributed reagents/materials/analysis tools, reviewed drafts of the paper.

Steven W. Salisbury conceived and designed the experiments, performed the experiments, analyzed the data, contributed reagents/materials/analysis tools, wrote the paper, reviewed drafts of the paper.

The following information was supplied regarding data availability:

Herne, Matthew; Tait, Alan; Weisbecker, Vera; Hall, Michael; Nair, Jay; Cleeland, Michael; Salisbury, Steven (2017): Victorian ornithopod NMV P221080: CT data of pes and tail. Figshare.

https://doi.org/10.6084/m9.figshare.5467990.v1

The following information was supplied regarding the registration of a newly described species:

Publication LSID: urn:lsid:zoobank.org:pub:0ACF3BE9-8E2F-4FEA-94B9-E418BE912418

Genus name: urn:lsid:zoobank.org:act:BB4925A8-A049-4569-9AF2-80B28E999279

Species name: urn:lsid:zoobank.org:act:9E1765D7-756F-4CF2-A005-EC0B0BE996BA

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
