# Peer review of "A new small-bodied ornithopod (Dinosauria, Ornithischia) from a deep, high-energy Early Cretaceous river of the Australian–Antarctic rift system"

_PeerJ, doi:10.7717/peerj.4113_

## Round 0.1 · original submission · Minor Revisions

This is a very thorough and well documented description for this proposed new ornithopod. The level of detail and care sets a high standard, and the description will undoubtedly be very informative for interested readers. The three reviewers have provided a number of specific comments for consideration during revision. I highlight two in particular below (although all should be addressed in some form):

1) As noted by two of the reviewers, you should give the utmost consideration to adding a formal phylogenetic analysis. Although the specimen is fragmentary, you should be able to at least get an approximate position for placement within the ornithopod evolutionary tree. This is particularly important because you make statements on the relationships of this new taxon relative to other Gondwanan taxa; as currently presented, they are basically unsupported. You could use some of the available ornithopod phylogenies, and perhaps add a few of the important new characters you highlight here. Note also the alternative placements proposed by Boyd for some of the "ornithopods" discussed here, outside of Ornithopoda, as noted by reviewer Poole.

2) The CT scans should be described in more detail; at a minimum, slice dimensions (beyond just thickness) such as pixel/voxel size should be provided. Although not mandatory, I would strongly urge you to consider repositing the data in an accessible location (e.g., a database such as MorphoSource), if permitted by institutional policy.

Reviewer 1 ·

Basic reporting

The ms is well writen and provides good new and relevant information on both the geology of the site and anatomical characteristics of the specimens. The ms is supported by relevant and complete references from the available literature.
I would recommend further elaboration and discussion about the phylogeny of the new taxon.
Figures of osteological elements should have the same standarized scale bars in all of them. I recommend using a single bar, without subdivisions, indicating the scale (1 mm, 1 cm, 10 cm, etc).

Experimental design

The new taxon should be supported within a phylogenetic context. There are several data matrices that the authors can use. The diagnosis based on authapomorpies is ok but it is important that combination of other characters be supported by a phylogenetical analysis.

Validity of the findings

As said above, a phylogenetic analysis of the new form within the Ornithopoda clade is mandatory. Some of the putative autapomorphies proposed by the authors could be unveiled as homoplasies or ambiguously distributed features.
Authors need to provide data sets of characters, with the new taxon properly scored, and a posible phylogenetical position of it.

Additional comments

The ms is an important contribution about the poorly known ornithopod diversity from Australia. I encourage the authors to work a little deeper on the phylogeny of the new taxon.

·

Basic reporting

The article is clearly written, professional and unambiguous it its language. The introduction and context for the article are sufficiently developed, and the appropriate literature is cited. The body of the description is very thorough, and the quality and number of figures exceeds normal requirements. The measurement raw data are shared (when possible), with exceptions where ratios are provided without the raw data. The CT scan data/3D models do not appear to be included in the submission. The paper is self-contained and does not contain extraneous information. Hypotheses are not clearly laid out or tested, but given that it is a taxonomic description this is not expected.

Experimental design

This represented original research, and falls within the Aims and Scope of the journal. The research questions are defined and of interest to the pertinent scientific community. The investigations performed were thorough and rigorous. The anatomical description is very detailed and of high technical standard. There are minor issues with the limited quantitative analyses performed. It would have been helpful to see the specimen/taxon placed in a cladistic context, and while this is not required its absence should at least be explained/justified. For the most part, documentation of the mythologies is sufficient for replication (and I was able to replicate several of the results), but more details on CT scanning an measurements would be useful.

Validity of the findings

The conclusions are clearly stated and backed up by the body of the paper. The data presented are accurate (with a few minor exceptions). Speculations do not over extend past those supported by the data.

Additional comments

General comments:

This is a very thorough anatomical treatment of interesting small-bodied ornithischian material from the Early Cretaceous of Australia. The anatomical description is very thorough, and I was surprised by the amount of description given the limited nature of the specimen. I am very happy with the separation of the description from the comparison (although this slips up in a few places - especially the end). The number and quality of the illustrations is excellent. It is a good contribution to the field and will be a very useful paper and resource.

While I do not doubt that this is a distinct and diagnosable taxon, I do have some concerns over the diagnosis. There are a large number of proposed autapomorphies for such limited material, and for material that is often not regarded are being overly diagnostic for related taxa. While some of these are autapomorphies, and many of them are strong and likely to be robust in future analyses (e.g., 2, 3, 7), the bulk of these features are not apomorphic. For these other features, a list of taxa that share this feature should be provided, and would be greatly beneficial. Some proposed autapomorphies (e.g., 4) are actually several features that are grouped together to form and ‘autapomorphy’. Rather than one large list, it would be more useful to break it down into two categories: firstly proposed autapomorphies and secondly other characters which can be used to differentiate from taxon XX. I do prefer the discrete autapomorphies over those based on relative proportions, especially when considering the effect of size/growth on these features.

While I appreciate the use of quantitative data to show differences in proportions between taxa (Figure 28, S5, S7, S8) I do not believe the simple comparisons of ratios is the most useful test. A ratio has two components, the numerator and denominator, but when two ratios are different you do not know if that is due to a difference in the numerator or denominator (or both). Likewise when two ratios are the same, you have no context of the absolute size of the elements. Since you need both the numerator and denominator to calculate a ratio, it is far more informative to simply plot these against each other in the graph. If you log both axes, any line with a slope is 1 represents a constant ratio. To illustrate this, I have taken the raw data (where available) from Table S3, and quickly plotted it up as series of ratios, and also in a bloxplot (colour-coded for taxa) (see attached). This allows for an examination of relative size (position along a line of slope = -1) and absolute size (position along a line of slope = 1) to be observed at the same time(I realized I switched the axes, but the point is the same). All points falling along the diagonal lines (slope of 1) in the figure will have the same ratio. Boiling down a biplot of numerator/denominator sets into a series of ratios is the mathematical equivalent of projecting the 2D data of a graph onto a 1D line (specifically a line with a slope of -1). While I appreciate the need to reduce the number of axes when that number is greater than three, our minds are very good at teasing out relationship in 2D space, and there is no reason to simplify these data into 1D space. Doing so removes a lot of the useful data, specifically absolute size, and which of the ratio components is changing. That being said, I do not this that this is a huge issue with the paper, it is just a better way to present the data, and can hopefully be incorporated.

The original data for the described specimen are provided (Tables 2-5) which is very good. Some of the data for the other taxa used for comparisons are presented (Table S2, S3, S4) which is great, but the majority of the data presented are not the raw data, but the ratio between two metrics. The raw data cannot be back-calculated from these ratios, and (as argued above) ratios have limited use compared to plotting up the raw data. Including the raw would both allow for better current analyses, and will serve the science better moving forward.

There are several errors in the tables, which are either math errors or typos (I can’t tell which). For example, the ratios for two of the seven taxa (Mattaburasaurus and Parksosaurus) in table S2 cannot be obtained from the supplied raw data. I only really checked out this table (because it has the most raw data), but a thorough check of the remaining math should be done.

The lack of cladistics analysis is confusing. I am not suggesting that one should always be required when a new taxon is described - there are several reasons why this may not be the case. However, if one is discussing potential relationships of the new animal in both the abstract and conclusion, it is reasonable to expect a cladistics framework, or a justification as to why this was not done. In this case I suspect the analysis will result in a low resolution, large polytomy, due to the nature of the material and poor character scoring in these anatomic regions in related taxa. If this is the case, explain why and it may help convince others to look for characters in these areas. If one is planned, indicate that as well.

As far as I can tell the CT data and/or resulting 3D digital models are not included in the supplementary materials. Inclusion of these data may be required to be consistent with PeerJ data sharing policies. I will leave that decision to the editor.

All in all, a great paper about an exciting specimen/taxon. I am looking forward to seeing it published.


Specific comments:

Throughout the document “zygopophysis”/”zygapophysis” are used seemingly interchangeably. There may a reason for this, but if not, use a consistent spelling.

Line 85 – “hyperextended” is this mean to indicate ‘elongate’?

Line 97 – the first usage of “Eric the Red West” should include the abbreviation ‘ETRW’ that is used in the rest of the paper

Line 130 - One minor omission to the methods is the details as to how the measurements were obtained. Where they obtained by calipers, tape, or digitally from the CT scans?

Line 135 – What voltage and amperage were the CT scans performed at?

Line 147 – the 11% intervertebral gap that was added needs more explanation. Presumably this was added to more accurate reconstruct the length lost due to intervertebral disks? Since the tail is still largely articulated, I have a hard time understanding how the disk lengths could be lost while retaining articulation (since the soft tissue is what is holding it all together).

Line 397 – It is unclear what is meant by a “linear” spinal process. Are these not all linear? Does this mean thin? Straight? Of constant thickness?

Line 445-446 – There is an apparent conflict. The first line suggests that the sutures are difficult to distinguish past Ca8, yet the second line discusses the position of the sutures on Ca10 and ca13?

Line 471 – “The centrum OF the referred…”

Line 487 – “… the anterior and posterior margins of the spinal processes are parallel AND OF CONSTANT ANTEROPOSTERIOR WIDTH…..” This this is intended meaning? If so it should be clarified more.

Line 488 – I do not understand the intended meaning of “The dorsal tips of these spinal processes are convex and their ventral tips, angular”

Line 490 – “Proximately narrow” do you mean “proximally constricted”

Line 501 – “linear” is this meant to indicate straight?

Line 512 – I think the incorrect figure is referred to here. Do you mean Figure 12?

Line 514/5 – Here is the first comparison in a so far comparison free description.

Line 565/5 – Other comparisons. I am not overly bothered by this, but is does break down the nice discrete sections you have established.

Line 599 – The term “restoration” is unclear. Would “digital 3D model” or similar work better?

Line 630 – “The distal (‘dorsal’) margin is obtuse” I am unclear what this is trying to say. How can the distal margin of the tarsal element also be the dorsal margin?

Line 681 – It is unclear what is meant by “less-derived ornithopods”. This is a vague and imprecise term. Do you mean ‘non-iguanodontian ornithopods”?

Line 819 - “length centrum” should be “centrum length”

Line 857 and 861 – “Parksosaurus warreni”

Line 865 – “C. browni”should be italicised.

Line 984 – two successive periods

Line 1193 – It is unclear what is meant by “linear along their length”. Does this mean straight?

Line 1275 – It may be helpful to end this section with a statement similar to… “Future cladistic analyses may help to test these hypotheses of relationships.

Line 1334 – The Anderson et al 1985 is a bit out of date. It may be good to also refer to the updated Campione et al 2014 (Meth in Ecol Evol) paper.

Figures 9, 10, 20, 21, 22, 23, 24, 25, 26, 27, 31A , 33 – Rather than simply describe these as “virtual” I would suggest “Digital 3D models” or similar. It is just a more specific description of what they are.

Figure 16 – Label “C” is in the wrong position. I think most people will be able to figure out that it pertains to the image on the left, but an arrow may help.

·

Basic reporting

no comment

Experimental design

no comment

Validity of the findings

A few speculative statements in the anatomical descriptions should be moved to the discussion or removed. These are discussed in my comments below.

Additional comments

This paper presents a thorough description of a specimen that is important in understanding the taxonomy of Australian ornithopods. I particularly appreciate the thorough description of the geological context (at both a macro scale and at the quarry level), and the many figures showing the details of the anatomy of this specimen.

The descriptions of the anatomy would be improved by remaining focused on features that can be directly observed and refraining from speculation about things like the location of muscles. This can be included in the discussion section of the paper, but clutters the description section. There are also some features that are discussed, notably the tprl and the abductor groove on MT II, that are not clear in the figures. The models created from CT scans have low resolution, and while these are still useful in showing certain features not visible in the photographs, small features are not clear in these images. There are still clear autapormorphies of this specimen that can be discussed: I would advise emphasizing these to clean up the description section and make it more focused.
Additionally, lists of proportions can clutter a description with superfluous information. An example here, from the description of pedal digit I-1: “The proximodistal length of pd I-1 is 47% that of mt I and 27% that of mt III. The transverse width of pd I-1 at its proximal end is 56% that of proximal pd II-1 and the dorsoplantar depth, 50% that of proximal pd II-1.” These measurements are all available in Table 5—unless a particular proportion is something that is diagnostic or will be highlighted in the discussion in comparison to other taxa, it need not be listed in the written description.

Based on the evidence presented, I’m unconvinced by the pathology section. I would want to see either close-up photos or histological sections of the affected areas—this is something where surface texture is important, so CT imagery is not a useful figure to demonstrate this. In short, I would like to see a more convincing figure demonstrating the pathology. Otherwise, I would simply remove this section.

In terms of phylogeny, it would be worth mentioning or considering the analysis of Boyd (2015), in which many of the taxa referred to in this text as basal ornithopods are in fact recovered as basal neornithischians outside of Ornithopoda. While you can certainly disagree with those findings and continue referring to these taxa as basal ornithopods, this alternative phylogeny should at least be mentioned.

On the topic of terminology, there is a mixture in this manuscript of the terms anterior/posterior and cranial/caudal. While either set of terms is fine, please pick one and stick with it for consistency and clarity. There also seems to be no abbreviation of genus names—after the first use of a binomial, the genus can be abbreviated.

When multiple references are cited, they should be put in chronological order.

Again, I applaud the authors’ thoroughness in describing and illustrating this specimen, and I look forward to the publication of the manuscript. Following is a list of specific comments and suggestions for improving the manuscript. Attached is a pdf containing further edits—these are simply minor grammar/spelling/punctuation errors.

Line 97: insert (ETRW) after first use of the full name “Eric the Red West”
Line 269-78: The Interpretation of the Anchor Sandstone says paleosol deposits overlie the lake deposits, but the caption of figure 5A says the Anchor Ss has “shallow lacustrine at top”.
Line 384: I realize this is a diagnosis, not a phylogenetic systematic analysis, but I think that breaking features down in a systematic way can be instructive. Features 4 and 5 are each three different features, with some overlap between them. Feature 4 deals with (a) shape of the spinal process, (b) length of the spinal process, and (c) orientation of the spinal process. If each of these features are unique and useful in diagnosing Diluvicursor, they should be listed separately. Feature 5 is about the height of the neural arch, which is tightly tied to length of the neural spine already covered in feature 4. You compare it here to three different measurements, but each of these can also vary between species (thus, they are separate characters). I would stick to one reference point, probably centrum height, as a comparison. Comparing the height of the neural arch to the length of the transverse process is a different feature, since that number is highly dependent on the length of the transverse processes (which is covered in feature 6). There are similar issues with feature 11.
Line 398-9: You need to be more specific in your shape descriptions. What, precisely, is the difference between “boot-shaped” and “hatchet-shaped”?
Line 578-82: The terms “hatchet-shaped” and “boot-shaped” aren’t defined here. It would be helpful to describe the discrete features here: are the proximal and distal edges of the hemal arch parallel? Does either edge diverge from the other? Is it symmetric or asymmetric? These are much more useful descriptions. Things like hatched-shaped and boot-shaped are fine as short-hand descriptions for continued discussion of the features, but they need to be defined in a more rigorous way at some point in the manuscript.
Line 613: “The fibula is anterolaterally positioned relative to the tibia.” But you note later that the fibula is displaced. Because of that, I would avoid discussing the relative positions of the tibia and fibula, unless there is other evidence of their anatomical relationship.
Line 614: You refer to a condyle on the fibula, but previously discussed malleoli on the tibia. If using the term malleolus, you should use this for the lateral malleolus of the fibula as well.
Line 630: “The distal (‘dorsal’) margin…” I don’t understand how a feature on the astragalus is both distal and dorsal. Inserting a sentence at the beginning of the description explaining the orientation you will use for your description of tarsal/metatarsal/phalangeal elements would be useful.
Line 656-7: “The metatarsus is compact, elongate and roughly cylindrical in shape.” I would describe this as an elliptical cylinder, as the cross section is oblong.
Line 660-5: This could all be shortened to something like: “Despite diagenetically imposed imbrication of the metatarsals, all elements are here described in life position.”
Line 675-7: “The M. extensor hallicus longus (e.g., White et al., 2016) that would have located within the extensor groove on the distal condyle, likely extended proximally along the medial margin of mt I and mt II.” This seems entirely speculative—I’m not sure what value it adds to the description.
Line 681-2: “…as in less-derived ornithopods”. Less-derived than what? I would use the term “basal ornithopods” here, or more technically, “basally-branching ornithopods”. But also see Boyd 2015: some of these taxa may not be ornithopods, and are better referred to as basally-branching neornithischians.
Line 728: “Adductor and abductor tendons could also have located within the medial and lateral grooves” (of mt III). This is rather speculative. In addition, interior digits are abducted and adducted by interosseous muscles and lumbricals. While these could be termed “abductors” and “adductors” based on function, the muscles themselves are not generally referred to this way.
Line 754-5: “The plantar portion of the cotyle is split in the axial direction (Figs 19A, 23B), which could be pathological.” Why not taphonomic? Is there any evidence either way?
Line 769-70: “Placement of Diluvicursor pickeringi in the Ornithischia is supported by the combination of a distally tapering mt IV…” Is this supposed to be distally curving? I don’t really see (nor is there a description of) it tapering.
Line 796-7: “The triangular intervertebral processes developed on the posterior-most caudal vertebrae of Diluvicursor pickeringi (Fig. 17) are similarly unique.” I’ve actually seen something similar in Gasparinisaura, specimen MCS-Pv 001 at Museo Cinco Saltos, which has a nearly complete and articulated tail. The intervertebral process is only present on two or three vertebrae, but they are quite similar to what is shown in this paper. To my knowledge, this specimen hasn’t yet been described, so I’ll leave it to the authors’ discretion whether to make this comparison or not, but I thought you might like to know about it—it certainly strengthens the case for a close relationship between Australian and South American ornithopods. (The ventral grooves on posterior caudal vertebrae are also present in this specimen.)
831,3,5: Spell out the full names of the lamina, at least the first place they are mentioned in the description, rather than sprl, tprl, and prsl.
839-41: “A prominent prsl on the caudal vertebrae of theropods, such as the abelisaurid Majungasaurus crenatissimus (O'Connor, 2007), suggests this feature is also plesiomorphic in dinosaurs, although variably expressed among taxa.” Or it arises frequently due to parallelism. Either way, it’s not diagnostic.
847-55: Again, I’ll refer to the undescribed specimen of Gasparinisaura MCS-Pv 001. While it has some hemal arches with rounded distal ends, just two vertebrae distal to this and still within the middle caudals are hemal arches with asymmetric triangular ends similar to those seen in NMV P185992/NMV P185993. It is important to remember in these discussions that the shape of hemal arches changes quite drastically based on position even within middle caudal vertebrae of one individual. Drawing conclusions about taxonomy or systematics based on these bones is dangerous unless comparisons can be made between fairly complete series.
885-8: “…suggesting that the position designated Ca 3 in Diluvicursor pickeringi is close to correct. It is reasonable to suggest that up to four caudal vertebrae could have been present on the Diluvicursor pickeringi holotype anterior to that designated Ca 1.” Based on what? How did you arrive at the number four?
968-70: “Similar features of the hallux in the early ornithischian Lesothosaurus diagnosticus (Thulborn, 1972), suggests this condition may be plesiomorphic for ornithopods.” While it’s true that Lesothosaurus has an “early” temporal range, in the context of describing its features as plesiomorphic, the better term here would be “basal” or “basally branching”.
970: “However, the halluces of the early ornithischians…” see above.
973-4: “indicating that the plesiomorphic condition of the hallux in ornithischians is presently not understood.” Well, what about outgroups? What would they indicate?
996-7: “…within a dorsolaterally oriented extensor groove (Figs 31A–B, S6E–F).” Figure 31 does not show an extensor groove on any of the specimens—or at least nothing is labelled as such. Figure 24A and 24F show the extensor groove clearly for D. pickeringi.
1020: “coerzeei” Should be coetzeei
1028: “Anti-mortem”: I think you mean antemortem?
1152-4: “a protuberance is also developed on each of the paired spinoprezygapophyseal lamina (sprl), between which, the transprezygapophyseal lamina (tprl) extends dorsally to both the neural canal and the prezygapophyses”. I don’t understand this description. The protuberances on the sprls appear to be dorsal to the prezygopophyses. How does the tprl then extend dorsally from the protuberance of the sprl to the prezygopophyses?
1267: “coerzeei” Should be coetzeei
1384-9: “Investigations on disparity in dental and cranial features between both co-occurring dinosaurian and ancient mammalian herbivores have addressed questions of habitat preferences within these groups…. However, the palaeoecological implications of cranial, dental and postcranial disparity between small-bodied ornithopods have yet to be investigated.” You say these disparities have been investigated in dinosaurs previously, but not in ornithopods. Maybe the first sentence should specify “some dinosaur groups” or “non-ornithopod dinosaurs”.

Figure 1: The color of water in figure 1C is very similar to that of land in figure 1A, and the background land color in 1C matches the water in 1B. This makes it difficult, at first glance, to discern where the coastline is in 1C. It’s overall a lovely figure, but consider making the color schemes consistent across all parts.
Figure 5A: I’m a little confused by the parenthetical “(top only)” in the description of the Anchor Sandstone. The description says it is a channel deposit with shallow lacustrine sediments at the top. But you have more than the shallow lacustrine sediments in your section. If you are trying to indicate that the site doesn’t include the full thickness of the Anchor Sandstone, this could be indicated by dashed arrows continuing downward in the schematic cross section.
5B: Just for the sake of clarity, it might be useful to change “intertidal zone” to “modern intertidal zone”, and “shore platform” to “modern shore platform”.
Figs 9, 10, 20: Perhaps this is a difference in Australian vs. American English, but describing a vertebra as “virtual” sounds to me as if there is something reconstructed or not quite real about it. I would describe this as a three-dimensional model or maybe “CT imaging of anterior caudal vertebra”.
Figure 10: “tprl, intraprezygapophyseal lamina” should be transprezygopophyseal lamina to match descriptions in the text.
Figure 19: The description of the views would be better if the elements were indicated, e.g., 19A: Tibia, fibula, and tarsals in anterior view, pes in plantomedial view. I would also argue that the tarsals seen in 19B are in something closer to a distal view than anterior.
Figure 20: “pd #, pedal digit number and phalanx position”—since you only point out metatarsals in this figure, why not use something like “mt” (or at least shorten the description to “pedal digit number”).
Figure 24: It would be nice if the parts of this figure were realigned so that MT I and its phalanges were aligned. They can remain as separately labelled parts of the figure, but aligning them such that a lateral view of MT I is next to a lateral view of the phalanges would make their relationship easier to understand.
Figure 30C: why is the schematic reflected while the photograph is not? If the purpose is to compare with the right limb of Diluvicursor, just reflect both the photo and schematic.
Figure 31: what view are these seen in?
Figure 33: As mentioned above, the CT imaging is not the best way to demonstrate pathologies. Consider a close-up photograph here instead.
Figure 35: “Extent of viscera (brown shading) ventral to hypaxial musculature (dashed lines) not shown.” If the cloaca extends this far posteriorly, it would be located more medially then ventrally (for a nice illustration of this in Alligator, see Mallison et al. 2015). Honestly, I think you could ignore viscera, since this is such a schematic view, and since this isn’t relevant for most of the tail. However, if you want to include viscera, they should be properly placed medial to the left and right hypaxial muscles.

Table 1: What sources are you following for this nomenclature? What do the asterisks mean?

Refernces
Boyd, C. A. (2015). The systematic relationships and biogeographic history of ornithischian dinosaurs. PeerJ, 3, e1523.
Mallison, Heinrich & Pittman, Michael & Schwarz, Daniela. (2015). PrePrint Version: Using crocodilian tails as models for dinosaur tails. PeerJ PrePrints. 3. e1653. 10.7287/peerj.preprints.1339v1.

---

## Round 0.2 · Minor Revisions

Thank you for your thorough response to the comments on the previous version of the manuscript. Everything is nearly ready to go in my opinion. My final request prior to acceptance is to provide the .tnt files for your phylogenetic analyses (in addition to the character scorings just for the new taxon). This would make reproduction of your phylogenetic results much easier for interested parties.

Once I have this final edit, I should be able to process the manuscript in very short order.

---

## Round 0.3 · accepted · Accept

Thank you for incorporating the requested changes.